# The reuniens nucleus of the thalamus facilitates hippocampo-cortical dialogue during sleep

Diellor Basha[1,2], Amirmohammad Azarmehri[3], Elian Proulx[1†], Sylvain Chauvette[2], Maryam Ghorbani[3], Igor Timofeev[1,2]*

[1]Département de psychiatrie et de neurosciences, Université Laval, Quebec, Canada; [2]CERVO Centre de recherche, Université Laval, Québec, Canada; [3]Department of Psychiatry and Neuroscience, Department of Electrical Engineering, Ferdowsi University of Mashhad, Mashhad, Iran

*For correspondence:
Igor.Timofeev@fmed.ulaval.ca

Present address: †CNS Diseases Research, Boehringer Ingelheim Pharma GmbH & Co. KG, Biberach an der Riß, Baden-Württemberg, Germany

Competing interest: The authors declare that no competing interests exist.

## eLife Assessment

The **important** manuscript presents **convincing** evidence of temporal correlations during specific oscillatory activity between the prefrontal cortex, thalamic nucleus reuniens, and the hippocampus, in naturally sleeping animals. Such correlations represent **solid** evidence to support the notion that the thalamic nucleus reuniens participates in the hippocampal and prefrontal cortex dialogue subserving memory processes.

**Abstract** Memory consolidation during sleep depends on the interregional coupling of slow waves, spindles, and sharp wave-ripples (SWRs), across the cortex, thalamus, and hippocampus. The reuniens nucleus of the thalamus, linking the medial prefrontal cortex (mPFC) and the hippocampus, may facilitate interregional coupling during sleep. To test this hypothesis, we used intracellular, extracellular unit and local field potential recordings in anesthetized and head restrained non-anesthetized cats as well as computational modelling. Electrical stimulation of the reuniens evoked both antidromic and orthodromic intracellular mPFC responses, consistent with bidirectional functional connectivity between mPFC, reuniens and hippocampus in anesthetized state. The major finding obtained from behaving animals is that at least during NREM sleep hippocampo-reuniens-mPFC form a functional loop. SWRs facilitate the triggering of thalamic spindles, which later reach neocortex. In return, transition to mPFC UP states increase the probability of hippocampal SWRs and later modulate spindle amplitude. During REM sleep hippocampal theta activity provides periodic locking of reuniens neuronal firing and strong crosscorrelation at LFP level, but the values of reuniens-mPFC crosscorrelation was relatively low and theta power at mPFC was low. The neural mass model of this network demonstrates that the strength of bidirectional hippocampo-thalamic connections determines the coupling of oscillations, suggesting a mechanistic link between synaptic weights and the propensity for interregional synchrony. Our results demonstrate the presence of functional connectivity in hippocampo-thalamo-cortical network, but the efficacy of this connectivity is modulated by behavioral state.

## Introduction

Interactions between the hippocampus and the medial prefrontal cortex (mPFC) are essential for cognition. During sleep, hippocampal and cortical activity is continually synchronized by cardinal sleep oscillations such as slow waves (SWs, 0.1–4 Hz), thalamocortical spindles (10–15 Hz), and hippocampal

sharp-wave ripples (SWRs, 100–300 Hz; *Isomura et al., 2006*; *Siapas and Wilson, 1998*). These rhythms have well-known roles in neuronal plasticity (*Chauvette et al., 2012*; *Diekelmann and Born, 2010*) and in the consolidation of hippocampus-dependent memories (*Maingret et al., 2016*). In particular, the hierarchical coupling of slow waves, spindles and SWRs is thought to play a key role in memory consolidation (*Diekelmann and Born, 2010*). Consistent with this hypothesis, the temporal coupling of hippocampal and mPFC sleep oscillations results in a selective increase in the recall of hippocampus-dependent memory (*Latchoumane et al., 2017*; *Maingret et al., 2016*).

At present, the pathways that coordinate the long-range coupling of slow waves, spindles and SWRs across the prefronto-hippocampal network are not well characterized. Although the hippocampus extends projections to several prefrontal areas (*Cavada et al., 1983*; *Goldman-Rakic et al., 1984*; *Irle and Markowitsch, 1982*; *Jay and Witter, 1991*; *Swanson, 1981*), few regions of the prefrontal cortex reciprocate (*Rajasethupathy et al., 2015*). With respect to medial regions of the prefrontal cortex, no direct projections to the hippocampus have been identified (*Beckstead, 1979*; *Goldman-Rakic et al., 1984*; *Reep et al., 1987*; *Room et al., 1985*; *Takagishi and Chiba, 1991*).

The nucleus reuniens of the thalamus (reuniens) is bidirectionally connected to the hippocampus and the mPFC (*Herkenham, 1978*; *Vertes et al., 2007*), making it an ideal hub for coupling hippocampo-cortical interactions during sleep. Reuniens is part of the ventral group of midline thalamic nuclei with widespread connections to cortical, thalamic, hypothalamic and brainstem structures (*Herkenham, 1978*; *McKenna and Vertes, 2004*; *Ohtake and Yamada, 1989*; *Robertson and Kaitz, 1981*; *Vertes et al., 2006*). Consistent with its anatomical connections to the hippocampus and the mPFC, reuniens plays a major role for memory function (*Ramanathan et al., 2018*; *Xu and Südhof, 2013*) and a growing body of evidence points its critical role in processing hippocampus-dependent memory (*Ramanathan and Maren, 2019*; *Ramanathan et al., 2018*). Although evidence for the functional significance of reuniens in cognitive operations is strong, little is known about the nature of reuniens-prefrontal interactions and their interplay with hippocampal events during sleep (*Di Prisco and Vertes, 2006*).

Considering the well-established link between sleep and memory consolidation (*Diekelmann and Born, 2010*; *Frankland and Bontempi, 2005*; *Timofeev and Chauvette, 2019*), we investigated the role of the reuniens in coupling hippocampo-cortical oscillations during non-rapid eye movement (NREM) and REM sleep. Using in vivo intracellular recordings, computational modelling, and spike/local field potential (LFP) recordings, we demonstrate that reuniens is driven by hippocampal oscillations during sleep and that it elicits fast, consistent synaptic responses in mPFC. Hippocampal and reuniens activity precedes the onset of mPFC spindles, suggesting a hippocampo-thalamic trigger for spindle initiation. In turn, the phase of mPFC slow waves determined the amplitude of reuniens spindles and of SWR timing, forming a functional feedback loop through the reuniens.

## Results

### Reuniens elicits fast, consistent synaptic responses in the mPFC

We obtained intracellular recordings of mPFC neurons in vivo (n=49 neurons) during targeted, electrical stimulation of the ventral hippocampus and reuniens from 16 anesthetized cats (*Figure 1A–D*). Stimulation of the hippocampus and reuniens elicited synaptic responses in 55% and 51% of recorded mPFC neurons, respectively. Hippocampal stimulation generated variable responses in the same neurons, consisting of complex sequences of either two excitatory postsynaptic potentials (EPSPs) followed by long-lasting hyperpolarization (*Figure 1E*, top, left), a depolarization-hyperpolarization sequence (*Figure 1E*, middle, left), or no response (*Figure 1E*, bottom, left). The polarity of early depolarizing responses elicited by hippocampal stimulation was dependent on membrane potential, reversing with progressive depolarization of the membrane at around –70 mV (*Figure 1E*, right), suggesting that the initial depolarization was an inhibitory postsynaptic potential (IPSP).

Reuniens stimulation elicited consistent EPSPs followed by long-lasting hyperpolarization (*Figure 1C*, bottom) at earlier latencies than hippocampal stimulation (6.05±3.6ms for reuniens and 9.95±4.6ms and hippocampal stimulation, respectively, t-test, p<0.01, *Figure 1G*). EPSPs elicited by reuniens stimulation had faster rising slopes compared to hippocampal stimulation (0.56±0.27 V/s for reuniens and 1.9±0.9 V/s for hippocampal stimulation, t-test, p<0.01) but were of lower amplitude (3.1±2.0 mV for reuniens and 8.4±4.1 mV for hippocampal stimulation, t-test, p<0.01). We observed

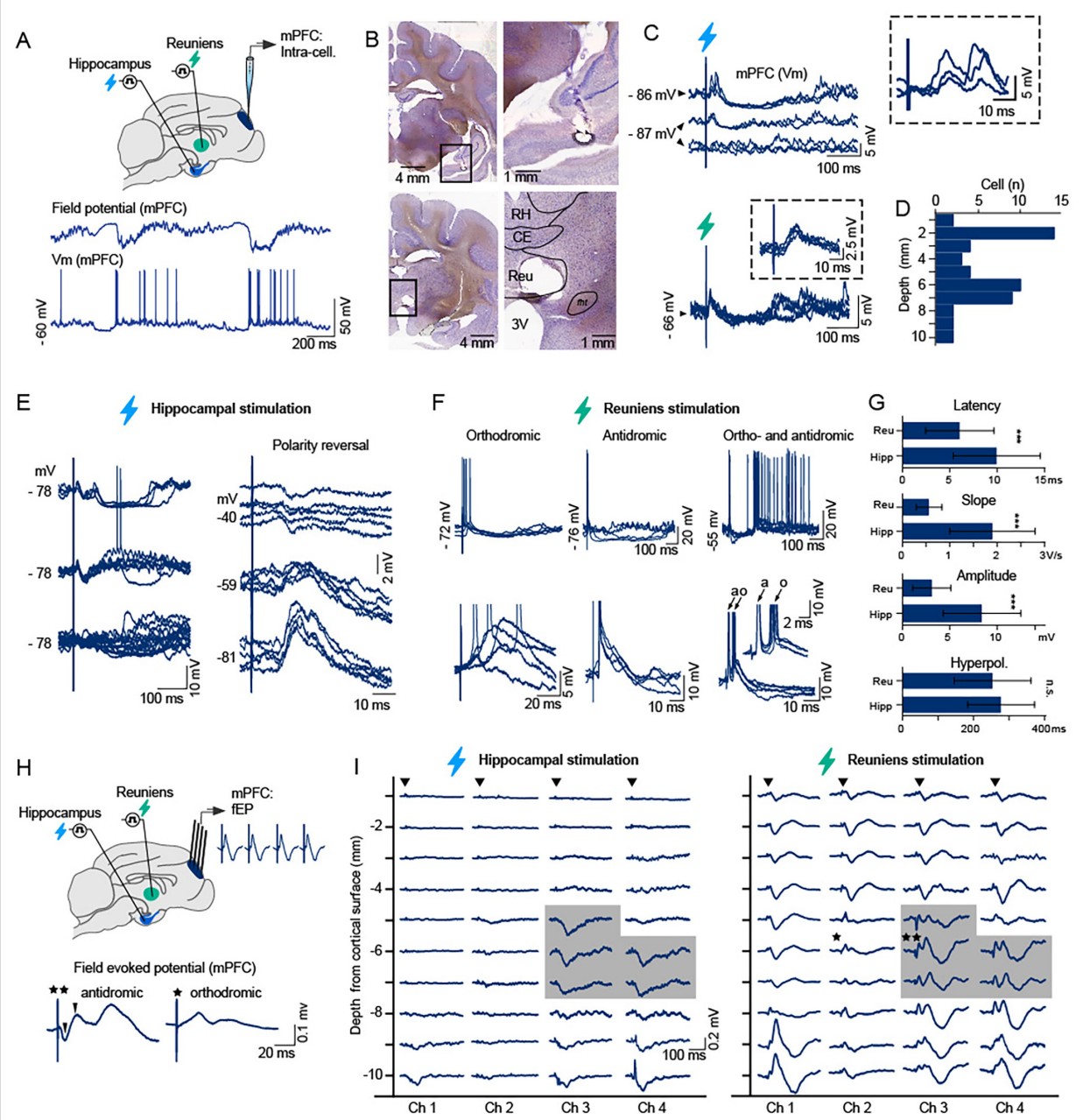

**Figure 1.** Differential synaptic responses of mPFC neurons to ventral hippocampal and reuniens stimulation. (**A**) Schematic of experimental setup: intracellular recordings of the mPFC were obtained in vivo during targeted stimulation of the ventral hippocampus and reuniens (top). Representative segment of intracellular recordings of the mPFC and proximal prefrontal LFP recorded in vivo during anesthesia (bottom). (**B**) Electrolytic lesions at stimulation sites in the ventral hippocampus (top) and midline thalamus (bottom) in Nissl-stained coronal sections. (**C**) mPFC synaptic responses to ventral hippocampal stimulation (top) and reuniens stimulation (bottom), recorded from same neuron, 5.2 mm ventral from the cortical surface. Note, variable, complex responses to hippocampal stimulation (top) and rapid, consistent responses to reuniens stimulation (bottom). (**D**) Location of recorded cells, estimated according to the dorsoventral position of the recording pipette with respect to the cortical surface. (**E**) Heterogeneous postsynaptic potential responses of a single mPFC neuron to consecutive ventral hippocampal stimulation (left): early double depolarizing response (top), early single response (middle), and no response (bottom). Reversal of early response with progressive depolarization (right), indicates involvement of GABA$_A$-mediated inhibition (n=5 pulses at each voltage). (**F**) Intracellular responses to single pulse stimulation of the reuniens nucleus were orthodromic (left, n=17 neurons), antidromic (middle, n=2 neurons) or both orthodromic and antidromic (right, n=1 neuron), consistent with bidirectional connectivity between the reuniens and the mPFC. Jitter in response latency characterized orthodromic spikes (bottom, left), compared to the consistent timing of antidromic spikes (bottom, middle) that were further confirmed with collision tests (not shown). (**G**) Population data (n=49 neurons from 15 acute animals, mean ± SD), depicting the response properties of mPFC neurons to ventral hippocampal and reuniens stimulation: response latency (6.05±3.6ms and 9.95±4.6ms for reuniens and hippocampal stimulation, respectively, t-test, p<0.01), response amplitude (3.1±2.0 mV and 8.4±4.1 mV

*Figure 1 continued on next page*

*Figure 1 continued*

for reuniens and hippocampal stimulation, respectively, t-test, p<0.01), slope of the early postsynaptic potential (0.56±0.27 V/s and 1.9±0.9 V/s for reuniens and hippocampal stimulation, respectively, t-test, p<0.01) and duration of the late, hyperpolarizing component (254±108ms and 278±94ms for reuniens and hippocampal stimulation, respectively, p=0.2, t-test). (**H**) Schematic of experimental setup illustrating targeted stimulation of the ventral hippocampus and reuniens and field responses recorded from the mPFC at 1 mm intervals along the medial wall of the pericruciate gyrus, prefrontal cortex (top). Evidence for antidromic field responses elicited by reuniens stimulation: short-latency, depth-negative events (bottom, left) compared to orthodromic responses lacking this event (bottom, right). (**I**) Topography of mPFC responses to hippocampal and reuniens stimulation. Stimulation of the reuniens evoked field responses in a wider prefrontal territory than ventral hippocampal stimulation. Traces show the average mPFC field response to 10 pulses (0.2ms, 0.3–1.5 mA at 1 Hz). Regions where ventral hippocampal stimulation elicited responses overlapped with regions where reuniens stimulation elicited antidromic responses (grey shading, stars indicate traces expanded in **H**, bottom). Ch1 was the most anterior electrode and was inserted at AP26. Channels were 1 mm apart in the antero-posterior axis. Data are presented as mean ± SD; *p<0.05, **p<0.01, ***p<0.001 (paired t-tests). *Reu - reuniens, RH – rhomboid, CE – central, pars medialis, fht – hypothalamic-tegmental fasciculus.*

no significant difference in the duration of the late, hyperpolarizing component in response to reuniens and hippocampal stimulations.

Reuniens stimulation elicited action potentials that were either orthodromic (85%), antidromic (10%) or both (5%), evidenced by jitter in the latency of orthodromic action potentials and consistent latency of antidromic action potentials (*Figure 1F*). Antidromic responses to reuniens stimulation, confirmed by collision tests, were recorded in cells located 6050–6600 µm ventral to the dorsal aspect of the cortical surface. The mean latency of antidromic spikes was 2.9±0.25ms (mean ± SD).

To characterize the prefrontal topography of reuniens and hippocampal receptive fields, we descended a 4-shank microelectrode array into the mPFC in 1 mm steps and recorded evoked field potentials in response to reuniens and hippocampal stimulation (*Figure 1H*). Reuniens stimulation evoked field postsynaptic potential in a wider prefrontal territory compared to hippocampal stimulation which was restricted to the ventroposterior part of the mPFC. In some locations of mPFC, early responses to reuniens stimulation (*Figure 1H, I*) corresponded to criteria of field potential antidromic responses (*Chang, 1953*). Areas in the prefrontal cortex that responded to hippocampal stimulation overlapped with areas exhibiting antidromic responses to reuniens stimulation (*Figure 1I*, shadowed area) which corresponds to area 32 of the prelimbic cortex. In the following experiments, we targeted this region of the mPFC.

## Reuniens and ventral hippocampal activity precede the onset of mPFC spindles

Sleep enhances communication in the hippocampo-cortical pathway by promoting interregional synchrony through highly rhythmic events such as slow waves and spindles. To investigate the role of the reuniens in mediating synchrony in the mPFC-hippocampal network, we performed LFP recordings of the mPFC and ventral hippocampus, as well as targeted spike/LFP recordings of the reuniens during natural sleep from four cats (*Figure 2A*, *Figure 2—figure supplement 1*). Recordings were segmented into wake, NREM and REM epochs, based on mPFC delta power, mPFC signal amplitude, and nuchal muscle activity (*Figure 2B–D*). During NREM sleep, LFP recordings and corresponding wavelet transforms of the signal suggested correlated activity among mPFC slow waves, spindles, hippocampal SWRs, and reuniens multi-unit activity (*Figure 2E*). We observed co-occurring reuniens and mPFC spindles (*Figure 2F*), characterized by significantly earlier onset of reuniens spindles relative to the onset of mPFC spindles (median = –0.083 s, n=49 sessions, *Figure 2G*).

NREM sleep analysis of reuniens single-unit activity relative to the phase of mPFC and reuniens spindle cycles revealed significant phase-locking of reuniens single-units to mPFC spindle cycles (n=24/49 single-units, mean ± SD = –80.87±41.64°, p=2.942 x $10^{-7}$, Rayleigh test, *Figure 2H*) and reuniens spindle cycles (n=29/49 single-units, mean ± SD = –118.25±47.72°, p=1.047 x $10^{-6}$, Rayleigh test *Figure 2I*). The mean phase preference of reuniens single-units was advanced for reuniens spindle cycles relative to mPFC spindle cycles. For pairs of co-occurring reuniens and mPFC spindles, spindle amplitudes were significantly correlated (*r*=0.54 using mPFC spindle as reference, p=3.22 × $10^{-27}$, *Figure 2J*, and *r*=0.47, p=8.61 × $10^{-25}$ when using Reuniens spindle as reference, *Figure 2K*).

Lastly, we investigated during NREM sleep the change in hippocampal and reuniens signal power during mPFC spindles relative to baseline, which demonstrated increased hippocampal ripple power and increased reuniens spindle power preceding the onset of spindles in the mPFC (*Figure 2L*, top).

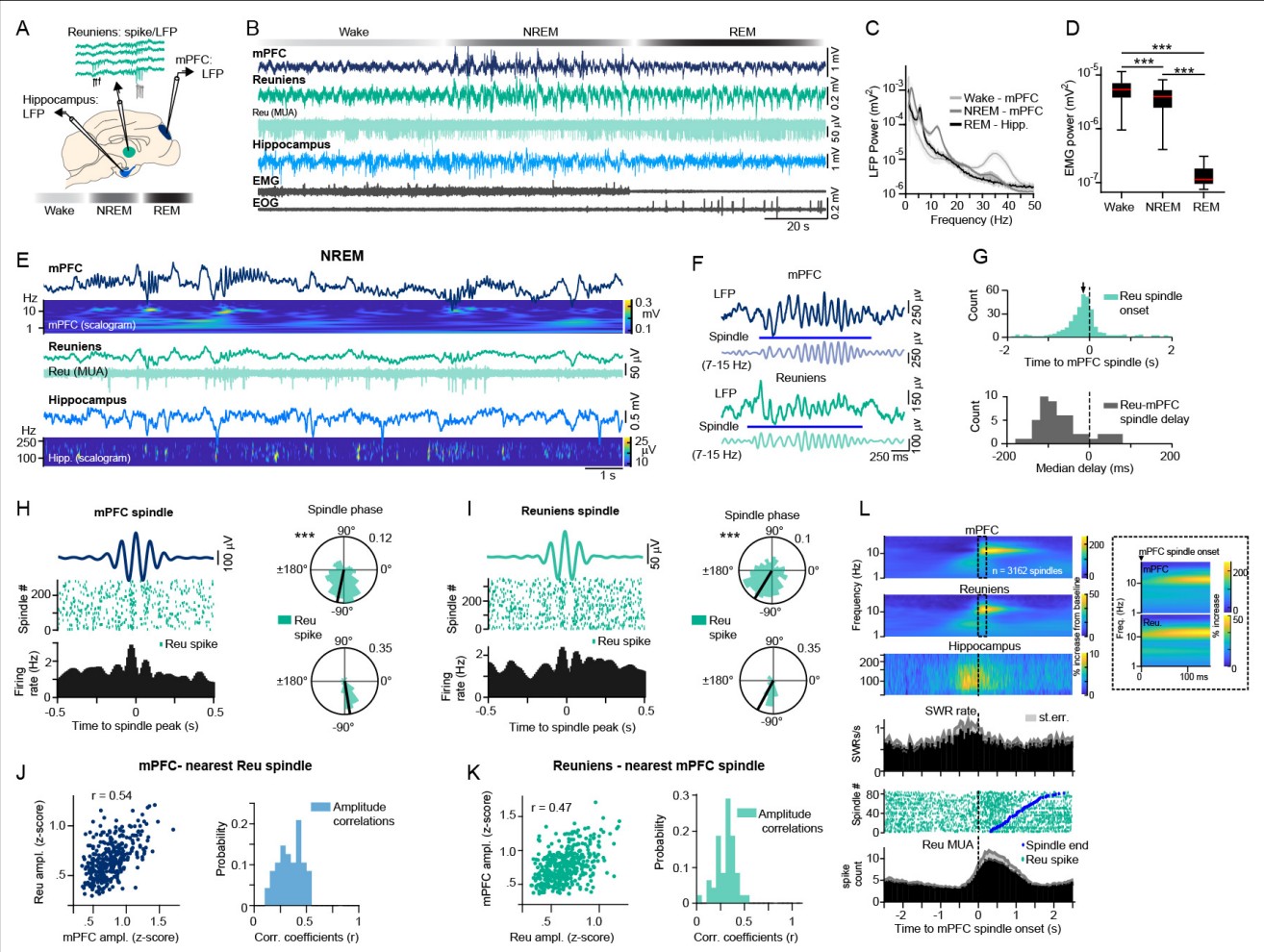

**Figure 2.** Reuniens activity and ventral hippocampal SWRs precede the onset of mPFC spindles. (**A**) Schematic of experimental setup illustrating LFP recordings of the ventral hippocampus and mPFC and targeted spike/LFP recordings of the reuniens during natural sleep. (**B**) A segment of LFP recordings of the mPFC, reuniens and ventral hippocampus during wake, NREM and REM states. For simplicity, we have chosen a segment with wake followed by unusually short NREM, followed by REM. Sleep/wake states were detected according to mPFC delta (1–4 Hz) power, mPFC signal amplitude and nuchal muscle activity (EMG). (**C**) The average power spectrum of LFP recordings in detected states, showing high gamma power in the mPFC during wake, high delta, and spindle power in the mPFC during NREM and high theta power in the ventral hippocampus during REM (n=52 recording sessions from 4 animals). (**D**) Muscle activity in detected wake, NREM and REM states. The red line in boxplot indicates the median, box limits represent the first and third quartile and whiskers indicate data range. ***p<0.001; *post-hoc* Tukey test after one-way ANOVA, n=52 recording sessions (four chronic animals). (**E**) Representative segment of LFP recordings during NREM with corresponding wavelet transforms of the signal, showing the correlated activity of mPFC slow waves, spindles, ventral hippocampal SWRs (top and bottom) and reuniens multi-unit activity (middle). The frequency axes for wavelet transforms are plotted in logarithmic scale. (**F**) Representative example of overlapping reuniens and mPFC spindles. Blue lines indicate automatically detected spindles in reuniens and mPFC LFPs. (**G**) The relative onset of reuniens spindles with respect to the onset of mPFC spindles in 100ms bins (top). The black arrow indicates the median delay time (median ± mean absolute deviation [MAD]=−0.147 ± 0.556 s, Wilcoxon signed rank test, p=6.13 x $10^{-15}$, n=346). The distribution of median delay times between each reuniens-mPFC spindle pair for each recording in 20ms bins (bottom, median = −0.083 s, n=49 sessions). (**H**) Reuniens single-unit activity relative to the phase of mPFC spindle cycles, aligned to mPFC spindle peaks. The average prefrontal LFP during spindles, filtered in the spindle frequency range (top left). Spike raster and the associated peri-event histogram of a representative reuniens single-unit during the above mPFC spindle (bottom left, 10ms bins). Phase-locking of the same reuniens single-unit to mPFC spindle cycles (top right, n=351 phases, p=9.120 x $10^{-29}$, Rayleigh test) and the mean phase preference of all significantly modulated reuniens single-units (bottom right, n=24/49 single-units from 4 chronic animals, mean ± SD = −80.87±41.64°, p=2.942 x $10^{-7}$, Rayleigh test). (**I**) Reuniens single-unit activity relative to the phase of reuniens spindle cycles. Same analysis as in H but referenced to reuniens spindles in one session (n=451 phases, p=8.048 x $10^{-18}$, Rayleigh test) and population data (n=29/49 single-units from 4 chronic animals, mean ± SD = −118.25±47.72°, p=1.047 x $10^{-6}$, Rayleigh test). Polar histograms indicate the normalized count of LFP phases at spike time. The black line indicates the mean phase angle of the associated histogram. Spikes occurred at earlier phases referenced to reuniens spindles compared to mPFC spindles (p=0.010, Watson-Williams test). (**J**) The amplitude of spindles detected in the mPFC versus the amplitude of the nearest reuniens spindle in one recording session (top, n=343 spindles, *r*=0.54, p=3.22 × $10^{-27}$, Pearson correlation) and the value of coefficients (**r**) in mPFC-reuniens amplitude correlations for all sessions (bottom, n=48 sessions

*Figure 2 continued on next page*

*Figure 2 continued*

with at least 50 spindles). (**K**) The amplitude of spindles detected in the reuniens versus the amplitude of the nearest mPFC spindle in one recording session (top, n=428 spindles, r=0.47, p=8.61 × 10⁻²⁵, Pearson correlation) and the value of coefficients (**r**) for reuniens-mPFC amplitude correlations for all sessions (n=48 sessions with at least 50 spindles). (**L**) The average change in LFP power during mPFC spindles (n=3162) relative to baseline, referenced to spindle onset, showing increased ventral hippocampal ripple power, and increased reuniens spindle power prior to the onset of spindles in the mPFC (top). Histogram of SWR rate in the same time window, referenced to the onset of the same mPFC spindles as above (middle). Compared to shuffled data, SWR rate was significantly higher in the 250ms window prior to the onset of mPFC spindles (t-test, p<0.05). Raster of reuniens multi-unit activity for a single recording session (n=86 spindles) and the population peri-event histogram (n=3,162 spindles), referenced to the onset of mPFC spindles. Grey in histograms represents standard error. See also *Figure 2—figure supplement 1* for unit detection.

The online version of this article includes the following figure supplement(s) for figure 2:

**Figure supplement 1.** Single-unit detection method from tetrode recordings.

The histogram of SWR incidence in the same time window, referenced to the onset of mPFC spindles, indicated increased SWR incidence prior to the onset of mPFC spindles (*Figure 2L*, middle). Reuniens multi-unit activity increased prior to the onset of mPFC spindles (*Figure 2L*, bottom).

## mPFC slow waves - spindle phase amplitude coupling in the reuniens-mPFC network

To examine the modulation of reuniens and hippocampal activity by mPFC slow waves, we analyzed the relationship between the phase of mPFC slow waves with respect to spindles and SWRs during NREM sleep. *Figure 3A* shows a representative example of a coupled slow wave-spindle event from simultaneous recordings of the mPFC and reuniens. We analyzed the average time-frequency representation of mPFC amplitudes locked to the peak of the mPFC slow wave (*Figure 3B*), as well as the average time-frequency representations of reuniens amplitudes, locked to the peak of the same mPFC slow waves (*Figure 3C*), which showed significant modulation of spindle amplitudes by the phase of the mPFC slow wave.

The strength of slow wave-spindle coupling, estimated by the absolute value of the synchronization index (SI) was higher for mPFC slow wave-reuniens spindle coupling compared to mPFC slow wave-mPFC spindle coupling (median ± mean absolute deviation [MAD], SI strength = 0.18 ± 0.09 and 0.24±0.11 for mPFC and reuniens spindles, respectively, p=2.126 × 10⁻¹⁷, Wilcoxon signed rank test, *Figure 3D*). A comparison of slow wave-spindle coupling strength between mPFC-mPFC pairs and mPFC-reuniens pairs is shown in *Figure 3E*. The amplitude of reuniens spindles increased at earlier phases of the mPFC slow wave compared to the amplitude of mPFC spindles, which increased at later phases of the mPFC slow wave (*Figure 3B, C and F*). This finding suggests that mPFC can trigger reuniens spindles, but spindles recorded in mPFC are not necessarily imposed by reuniens spindles.

Next, we examined the relative delays in the timing of mPFC and reuniens slow waves during NREM sleep. *Figure 3G* shows a representative example of an overlapping mPFC and reuniens slow wave, with circles indicating slow wave peaks. Single session data in *Figure 3H* demonstrates the delay between reuniens and mPFC slow waves, showing significant lag in the time of reuniens slow waves relative to mPFC slow waves (median ±MAD = 0.126±0.043 s, p=3.66 × 10⁻¹⁵, n=82, Wilcoxon signed rank test). The distribution of median delay times for all recording sessions between each reuniens-mPFC slow wave pair are shown in *Figure 3I*.

Lastly, we analyzed the phase-locking of ventral hippocampal SWRs to mPFC slow waves. *Figure 3J* shows a representative example of a coupled slow wave-SWR event. The distribution of slow wave phases at the time of SWRs was non-uniform, indicating significant phase-locking of SWRs to mPFC slow waves at a mean phase of 162.7±29.37° (p=5.02 × 10⁻⁶, Rayleigh test, *Figure 3K*). We found no significant phase-locking of SWRs to spindle cycles in the majority of recording sessions (n=3/51 with significant phase for mPFC spindle-SWR and n=10/51 sessions with significant phase for reuniens spindle-SWR coupling, *Figure 3—figure supplement 1* and *Table 1*).

## Hippocampal oscillations modulate reuniens activity

Considering the comodulation of reuniens and hippocampal activity prior to mPFC spindles, we asked whether hippocampal oscillations directly influence firing activity in the reuniens. A representative example of LFP recordings of the ventral hippocampus, mPFC, and reuniens during REM sleep is shown in *Figure 4A*. The wavelet spectrogram of the hippocampal trace demonstrates prominent

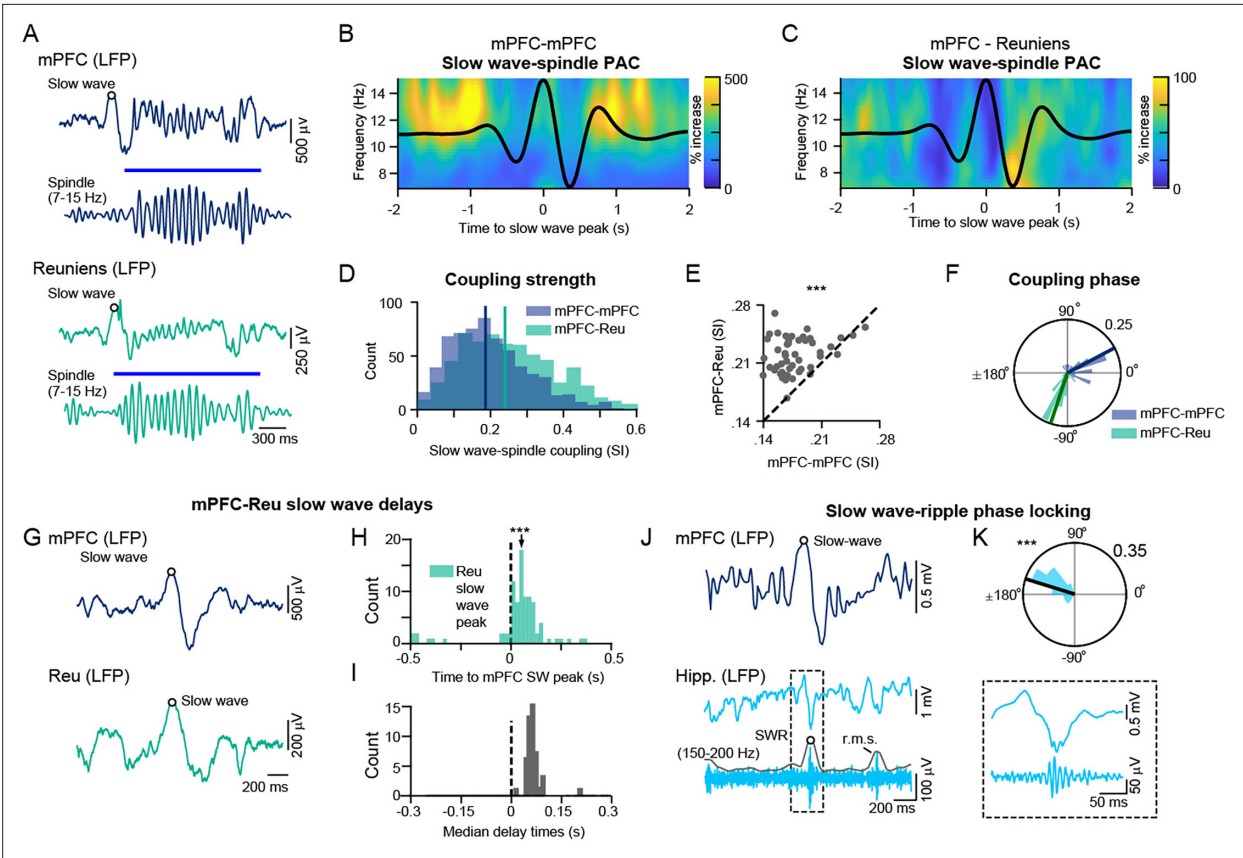

**Figure 3.** Slow wave-spindle and slow wave-SWR coupling in the mPFC-reuniens-hippocampal network. (**A**) A single trial of a slow wave-spindle event, showing raw and filtered for spindle LFP traces of the mPFC (top) and reuniens (bottom). (**B**) Phase-amplitude coupling (PAC) of mPFC spindles and mPFC slow waves, shown as the average time-frequency representation of mPFC amplitudes in the spindle frequency range, locked to the peak of the mPFC slow wave (percentage change from pre-event baseline, n=626 slow waves). (**C**) Phase-amplitude coupling of reuniens spindles and mPFC slow waves, shown as the average time-frequency representations of reuniens amplitudes, locked to the peak of the same mPFC slow waves as in B (percentage change from pre-event baseline, n=626 slow waves). Black curves in B and C show the average of the mPFC LFP, filtered in the slow wave frequency range (0.5–2 Hz). (**D**) Histogram of slow wave–spindle coupling strengths for mPFC slow waves-mPFC spindles (blue) and mPFC slow wave-reuniens spindles (green). Vertical lines indicate the median of the distributions (median ±MAD, SI strength = 0.18 ± 0.09 and 0.24±0.11 for mPFC and reuniens spindles, respectively, p=2.126 x 10$^{-17}$, Wilcoxon signed rank test). (**E**) Slow wave-spindle coupling strength of mPFC-mPFC pairs versus mPFC-reuniens pairs (n=49 sessions, p=3.39 x 10$^{-2}$, Wilcoxon signed rank test). Each circle represents one session. (**F**) Population data showing the relative preference of spindles to the phase of the mPFC slow wave, shown as a histogram of SI angle means for mPFC slow wave- mPFC spindle coupling and mPFC slow wave-reuniens spindle coupling (mean ± SD = 27.92±37.59°, p=4.661 x 10$^{-9}$, n=26 and mean ± SD = –108.48±21.17°, p=1.278 x 10$^{-20}$, n=36, for mPFC and reuniens respectively, Rayleigh test). (**G**) A single trial of an overlapping slow waves, showing raw and filtered LFP traces of the mPFC (top) and reuniens (bottom). Circles indicate slow wave peaks. (**H**) Single session data, showing the distribution of delays between reuniens and mPFC slow waves (vertical dashed line). Reuniens slow waves lagged mPFC slow waves (median ±MAD = 0.126±0.043 s, p=3.66 x 10$^{-15}$, n=82, Wilcoxon signed rank test). (**I**) Population data of median delay times between each reuniens-mPFC slow wave pair (n=49 sessions, median of the distribution = 0.136 s). (**J**) Single-trial of raw mPFC and raw and filtered for ripples ventral hippocampal LFPs during one overlapping slow wave-SWR event. Inset shows an expanded view of the detected SWR. (**K**) Circular histogram of mPFC slow wave phases at the time of SWRs (mean phase ± SD = 162.7±29.37°, p=5.02 x 10$^{-6}$, Rayleigh test). *p<0.05, **p<0.01, ***p<0.001. See also ***Figure 3—figure supplement 1*** for spindle-SWR coupling.

The online version of this article includes the following figure supplement(s) for figure 3:

**Figure supplement 1.** Spindle-SWR phase locking.

theta-band activity during REM with periodic locking of reuniens multi-unit activity to hippocampal theta cycles (***Figure 4B***). Analysis of reuniens firing around the peak of hippocampal theta cycles, shows significant phase-locking of reuniens single-unit activity to the theta rhythm during REM (***Figure 4C–F***).

Power spectra of mPFC, reuniens, and hippocampal signals during REM are characterized by prominent peaks in the theta range, with coherence between mPFC-hippocampus, reuniens-hippocampus, and mPFC-reuniens recording pairs in the theta range during REM (***Figure 4G and H***). mPFC-reuniens

**Table 1.** Spindle-SWR coupling, p-values per session.

| Recording Session | Recording Duration (s) | Sleep Duration (s) | mPFC-hipp **p-val.** | Reu-hipp **p-val.** |
|---|---|---|---|---|
| Amande_Day6S2 | 5888 | 1460 | 0.273 | 0.786 |
| Amande_Day7S1 | 8210 | 720 | 0.726 | 0.718 |
| Amande_Day7S2 | 3573 | 2180 | 0.74 | 0.204 |
| Amande_Day8S1 | 8343 | 720 | 0.202 | **0.022** |
| Amande_Day8S2 | 5734 | 1900 | 0.505 | 0.927 |
| Amande_Day8S3 | 5693 | 1420 | 0.65 | 0.725 |
| Amande_Day9S1 | 4543 | 660 | **0.039** | 0.875 |
| Amande_Day9S2 | 4413 | 1560 | 0.757 | 0.866 |
| Amande_Day9S3 | 4943 | 3640 | 0.964 | 0.066 |
| Amande_Day10S1 | 3163 | 240 | 0.318 | 0.772 |
| Amande_Day10S2 | 6488 | 4560 | 0.767 | 0.152 |
| Amande_Day11S1 | 3033 | 440 | 0.068 | 0.429 |
| Amande_Day11S2 | 9016 | 4710 | 0.594 | **0.019** |
| | | | | |
| Cocotte_Day4S1 | 6763 | 1900 | 0.155 | 0.034 |
| Cocotte_Day4S2 | 4033 | 1000 | 0.081 | 0.505 |
| Cocotte_Day5S1 | 6423 | 1720 | 0.975 | 0.427 |
| Cocotte_Day6S1 | 6043 | 580 | 0.72 | **0.013** |
| Cocotte_Day6S2 | 6033 | 1780 | 0.429 | 0.571 |
| Cocotte_Day7S1 | 5965 | 2160 | 0.494 | 0.681 |
| Cocotte_Day7S2 | 4190 | 1580 | 0.617 | 0.78 |
| Cocotte_Day7S3 | 5495 | 2700 | 0.91 | 0.139 |
| Cocotte_Day8S1 | 8389 | 5480 | 0.449 | **6.36E-04** |
| Cocotte_Day8S2 | 6578 | 4700 | 0.476 | **0.019** |
| Cocotte_Day8S3 | 2588 | 720 | 0.798 | 0.619 |
| Cocotte_Day9S1 | 10073 | 6520 | 0.937 | 0.174 |
| Cocotte_Day9S2 | 5289 | 2740 | 0.581 | **0.046** |
| Cocotte_Day10S1 | 4583 | 480 | 0.142 | 0.957 |
| Cocotte_Day10S2 | 5953 | 4100 | 0.358 | 0.624 |
| Cocotte_Day10S3 | 6018 | 2280 | 0.667 | 0.386 |
| Cocotte_Day11S1 | 14023 | 9680 | 0.055 | 0.104 |
| Cocotte_Day12S1 | 9863 | 6840 | 0.646 | 0.928 |
| | | | | |
| Garfield_Day4S2 | 3428 | 940 | 0.187 | 0.234 |
| Garfield_Day4S3 | 4753 | 1720 | **0.003** | 0.013 |
| Garfield_Day5S1 | 8439 | 1920 | 0.737 | 0.345 |
| Garfield_Day5S2 | 6464 | 2680 | 0.123 | 0.557 |
| Garfield_Day6S1 | 8073 | 2080 | 0.396 | 0.781 |
| Garfield_Day6S2 | 7073 | 4400 | 0.065 | 0.895 |

*Table 1 continued on next page*

*Table 1 continued*

| Recording Session | Recording Duration (s) | Sleep Duration (s) | mPFC-hipp **p-val.** | Reu-hipp **p-val.** |
|---|---|---|---|---|
| Garfield_Day7S1 | 8623 | 2700 | 0.619 | 0.494 |
| Garfield_Day7S2 | 6664 | 4360 | 0.277 | **0.002** |
| Garfield_Day8S1 | 4884 | 340 | 0.775 | 0.929 |
| Garfield_Day8S2 | 5468 | 2520 | 0.556 | 0.237 |
| Garfield_Day8S3 | 4438 | 3020 | 0.497 | 0.319 |
| Garfield_Day9S1 | 4438 | 2980 | 0.3 | 0.248 |
| Garfield_Day9S2 | 4568 | 1780 | 0.321 | 0.43 |
| Garfield_Day9S3 | 6329 | 3680 | **0.029** | **0.013** |
| Garfield_Day10S1 | 7068 | 840 | 0.719 | 0.409 |
| Garfield_Day10S2 | 1089 | 120 | 0.459 | 0.135 |
| Garfield_Day11S1 | 6684 | 1980 | 0.857 | 0.258 |
| Garfield_Day12S1 | 5893 | 1520 | 0.538 | 0.971 |
| Garfield_Day12S2 | 7503 | 3360 | 0.633 | 0.701 |
| Garfield_Day13S1 | 3063 | 300 | 0.296 | 0.466 |

Bold indicates significant values.

recordings were further comodulated by a slow, irregular potential observed in *Figure 4A* and in the broad low frequency peak (*Figure 4H*, bottom). Cross-correlations and relative time lags of reuniens-hippocampus and mPFC-hippocampus recording pairs during REM are illustrated in *Figure 4I*. mPFC signals lagged hippocampal signals by a longer delay than reuniens signals lagged hippocampus (mean lag ± SD = 87.8±55.5ms and 109.1±41.52 for reuniens and mPFC, respectively, n=20 sessions, p=0.002, t-test, *Figure 4J*, top) and peak cross-correlation coefficients were significantly higher for reuniens-hippocampus recording pairs than for reuniens-mPFC (mean peak value ± SD = 0.44±0.13 and 0.22±0.07 for reuniens and mPFC respectively = 20 sessions, t-test, $p<10^{-20}$, *Figure 4J*, bottom).

Next, we analyzed reuniens firing activity during SWRs in NREM. *Figure 4K* shows representative LFP recordings of the hippocampus and reuniens multiunit activity during SWR events in NREM sleep. In 20.4% of reuniens single-units (10/49), firing rate increased significantly following the onset of SWRs in the hippocampus, compared to shuffled events in the same period (*Figure 4L*, 150 ms post-SWR onset). Firing rates decreased significantly in 32.6% (16/49) of single-units following SWR onset. The average thalamic LFP around the onset of the SWR was characterized by a negative, 250 μV deflection coincident with increased reuniens firing.

## Hippocampal-thalamocortical model

To explore the parameters that control interregional synchrony during NREM sleep, we constructed a neural mass firing rate model consisting of two cortical networks (N1 and 2), representing mPFC layers 5 and 6, respectively, and two hippocampal (CA1-CA3) and three thalamic networks (TRN, MD, and Reu, *Figure 5A*). Sharp waves and slow waves are spontaneously generated in the isolated CA3 and cortical networks respectively as a result of adaptation and excitatory-to-excitatory recurrent connections. SWRs are reproduced in the CA1 network during the active phase of the sharp wave due to recurrent connections between the CA1 excitatory and inhibitory neurons with the firing rates sharply dependent on the membrane potential in this firing rate model (*Azimi et al., 2021*). The oscillations in the range of spindle frequency are spontaneously generated in the bursting, recurrently connected MD-TRN network. The beginning and end of the spindles are controlled by cortical inputs in the thalamocortical model with long-range bidirectional projections between cortical and thalamic networks. In the full hippocampo-cortico-thalamic model, the CA1 is directly connected to the cortical network while reuniens, which is bidirectionally connected to both CA1 and cortical networks, mediates the

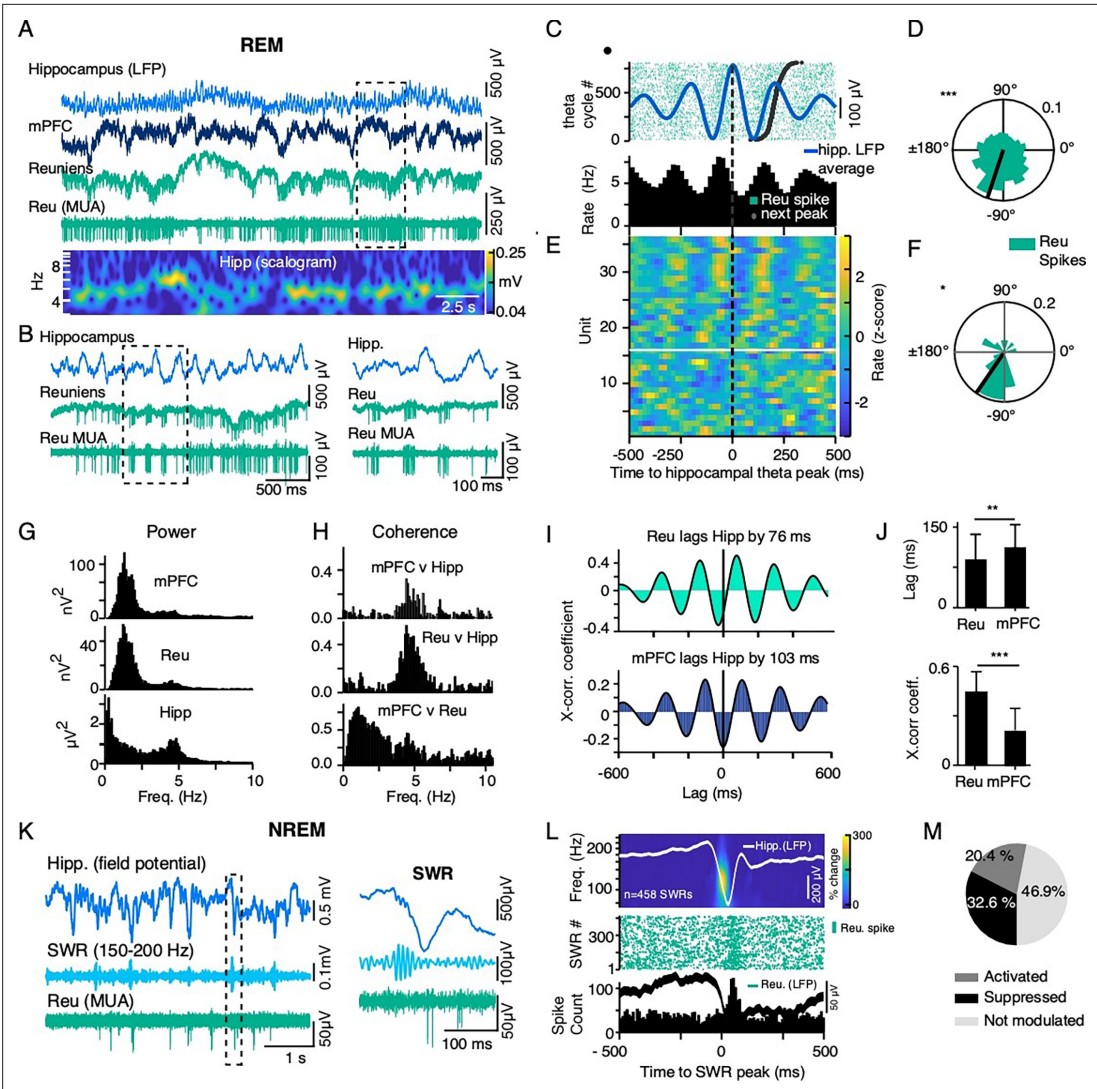

**Figure 4.** Ventral hippocampal theta oscillations and SWRs modulate reuniens activity. (**A**) Representative example of LFP recordings of the ventral hippocampus, mPFC and reuniens during REM sleep (top three traces) and the bandpassed (300–8000 Hz) trace of the reuniens signal showing multiunit activity. Wavelet spectrogram of the hippocampal trace, showing prominent theta-band activity during REM. The frequency axis is logarithmic. (**B**) Expanded view of the recording epoch in the dotted-line box in panel A (left), further expanded in time (right), showing occasional locking of reuniens multi-unit activity to ventral hippocampal theta cycles. (**C**) Peri-event histogram of a representative reuniens single-unit, referenced to the peak of ventral hippocampal theta cycles (top). The raster is ordered according to the peak-to-peak latency, with short periods at the bottom and long periods at the top. Dark black dots indicate the timing of the next peak in the trial. The light blue trace is the average waveform of the hippocampal LFP. (**D**) Circular histogram of theta phase at the time of reuniens spikes, showing significant phase-locking of a reuniens single-unit activity to ventral hippocampal theta cycles (same single-unit as in C, n=5,207 spikes, p<10$^{-20}$, Rayleigh test). (**E**) Population data, showing peri-event histograms referenced to the peak of ventral hippocampal theta cycles as in B (n=36 single units). Units above the white line were significantly modulated by hippocampal theta cycles. (**F**) Population data, showing the circular distribution of reuniens firing relative to theta cycles for all modulated single units (n=21/36 single-units, mean ± SD = –124.5±59.8°, p=0.011, Rayleigh test). (**G**) Single session data, showing power spectra of mPFC, reuniens and ventral hippocampal signals during REM with prominent peaks in the theta range. (**H**) Single session data, showing signal coherence between mPFC-hippocampus, reuniens-hippocampus and mPFC-reuniens recording pairs during REM. (**I**) Single session data, showing cross-correlations and relative time lags of reuniens-hippocampus and mPFC-hippocampus recording pairs during REM. (**J**) mPFC signals lagged ventral hippocampal theta signal by longer delay than reuniens signals lagged hippocampus (top, mean lag ± SD = 87.8±55.5ms and 109.1±41.52 for reuniens and mPFC, respectively, n=20 sessions, p=0.002, t-test). Peak cross-correlation coefficients were significantly higher for reuniens-Hippocampus recording pairs than for reuniens-mPFC (bottom, mean peak value ± SD = 0.44±0.13 and 0.22±0.07 for reuniens and mPFC respectively = 20 sessions, t-test, p<10$^{-20}$). (**K**) Representative LFP recordings of the ventral hippocampus and of reuniens multiunit activity during NREM sleep. Light blue traces show the bandpass filtered hippocampal recording in the ripple band (150–200 Hz). Light green shows multiunit activity in reuniens during SWR. Expanded view of a single SWR from the recording epoch in the dotted-line box and multiunit activity in reuniens. (**L**) Peri-event analysis of reuniens spiking activity around SWRs in one

*Figure 4 continued on next page*

*Figure 4 continued*

recording session. The time frequency representation shows the mean wavelet transform of 458 detected SWRs, estimated by Morlet wavelets (top). Overlayed traces show the grand average LFP in the ventral hippocampus (white) and reuniens (green), referenced to the onset of the SWR. Raster of reuniens spikes (every 10th sweep) and associated peri-event spike histogram, referenced to the onset of the SWR (bottom). (**M**) Proportion of reuniens single-units modulated by SWRs (10/49 activated, 16/49 suppressed). *p<0.05, **p<0.01, ***p<0.001.

cortical to CA1 influence. With large enough strength of multi-regional projections, the model can spontaneously reproduce the experimentally observed coupling between the slow waves, spindles, and SWRs (*Figure 5B*). We found that reuniens slow waves occur significantly later than cortical slow waves (median ± MAD = 0.113±0.019 s, p<10$^{-20}$, Wilcoxon signed rank test, *Figure 5C*). In contrast, thalamic spindles occur significantly earlier than cortical spindles (median ± MAD = –0.102±0.387 s, p=1.30 × 10$^{-7}$, Wilcoxon signed rank test, *Figure 5D*).

*Figure 5E* shows the average of cortical slow wave trough-locked time-frequency representations of the membrane potential of mPFC excitatory neurons and reuniens neurons. The slow wave-spindle coupling angles are nonuniformly distributed for both cortical and reuniens spindles indicating strong coupling with slow waves (mean ± SD = 148.7±23.9°, p<10$^{-20}$ and 27.4±20.4°, p<10$^{-20}$, for mPFC and reuniens respectively, Rayleigh test). In quantitative match to the experimental data, reuniens spindles occur mainly around the slow wave peak (active phase) while cortical spindles mainly occur around the slow wave trough. The slow wave-spindle coupling strength is also higher for reuniens than cortical spindles median ±MAD SI strength = 0.58 ± 0.11 and 0.66±0.15 for mPFC and reuniens, respectively, p=8.45 × 10$^{-9}$, Wilcoxon signed rank test, (*Figure 5F*). Slow waves in the thalamocortical network are also accompanied by large negative deflections and a subsequent peak in CA1 network (*Figure 5G*). Phase locking analysis revealed that, consistent with the experimental data, SWRs are phase-locked to slow waves, occurring mainly before slow wave peaks (active phase, mean ± SD = –24.97±62.01°, p<10$^{-20}$, Rayleigh test, (*Figure 5H*). The bidirectional projections between reuniens and CA1 coordinate the communication between these two networks so that the membrane potential of the reuniens neurons increases after SWRs (*Figure 5I*). Reuniens spindles were also preceded by an increase in the SWR rate (*Figure 5J*), the peak SWR rate occurs 0.32 s before the spindle onset.

## Bidirectional projections between the reuniens and CA1 control multiregional interactions

To investigate how bidirectional projections between the reuniens and CA1 control multiregional interactions in the model, we explored hippocampal-thalamocortical coupling during sleep rhythms as a function of the strength of bidirectional projections between reuniens and CA1. The correlation between reuniens and CA1 membrane potentials during SWRs increased by increasing the strength of CA1 to reuniens projections (*Figure 6A*). In addition, the SWR activity before spindle onset is an increasing function of bidirectional projections between CA1 and reuniens (*Figure 6B*). Next, we studied the effect of the strength of reuniens-CA1 bidirectional projections on CA1 activity during cortical slow waves. The peak-to-trough magnitude of CA1 deflections around the cortical slow wave also increases by increasing the reuniens-CA1 bidirectional projections (*Figure 6C*). The slow wave mean phase at the time of SWR occurrence decreased by increasing the strength of the bidirectional projections between reuniens and CA1 so that SWRs mostly occur before and after slow wave peak for strong and weak projections, respectively (*Figure 6D*). These results suggest that for strong reuniens-CA1 projection, the depolarizing input from reuniens during down-to-up transition is large enough to induce the SWRs in CA1 before the slow wave peak. Further, slow wave phases at the time of SWR occurrence are more nonuniformly distributed for stronger CA1-reuniens projection, indicating stronger phase locking of SWRs to slow waves.

## Discussion

This study examined the functional relationship between mPFC, reuniens, and hippocampal activity in cats using electrophysiological and computational approaches. We found that reuniens activity during natural sleep was strongly modulated by hippocampal oscillations, and that reuniens-mPFC spindles were preceded by increased probability of hippocampal SWRs and ripple power. Electrical stimulation of reuniens generated orthodromic and antidromic spiking in mPFC neurons, indicating bidirectional

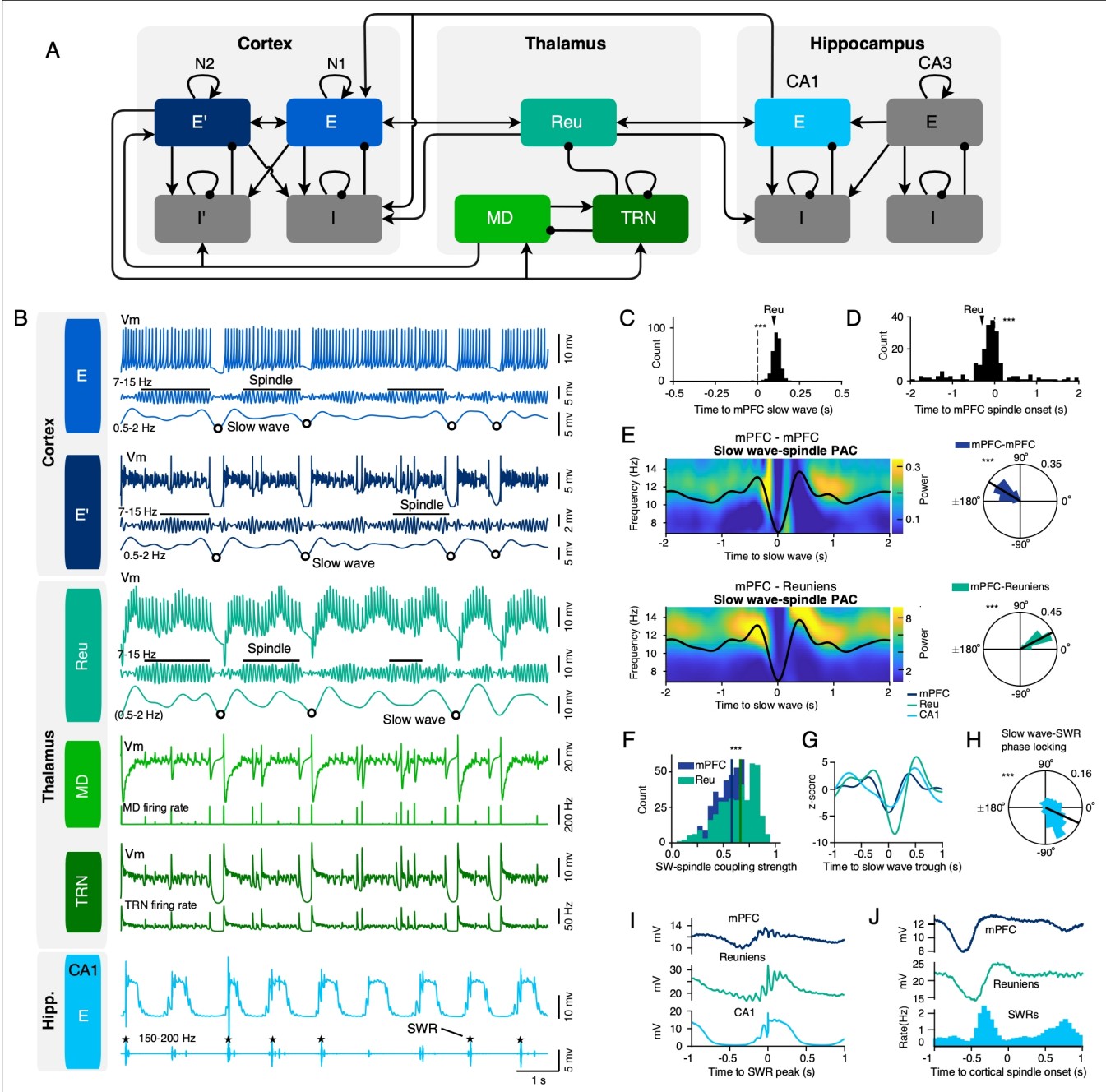

**Figure 5.** Multiregional temporal interactions during NREM sleep events in a hippocampal-thalamocortical model. (**A**) The model is composed of two cortical networks (N1 and 2, representing mPFC layers 5 and 6 respectively), two hippocampal (CA1-CA3) networks and three thalamic nuclei. The three thalamic networks represent the nucleus reuniens (Reu), mediodorsal thalamic nucleus (MD) and thalamic reticular nucleus (TRN). The cortical and hippocampal networks are composed of one group of identical excitatory (**E**) and one group of identical inhibitory (**I**) neurons. Excitatory synaptic connections are shown by arrows and inhibitory connections are indicated as lines ending in a dot. (**B**) From top to bottom: broadband and filtered (7–15 Hz, 0.5–2 Hz) traces of the membrane potential of excitatory neurons of cortical network 1 (light blue) and 2 (dark blue) with detected spindles (black horizontal lines) and slow waves (circles); broadband and filtered (7–15 Hz, 0.5–2 Hz) traces of the membrane potential of reuniens neurons (blue-green) with detected spindles and slow waves; broadband trace of the membrane potential and firing rate of MD (light green) and reticular neurons (dark green); broadband and filtered (150–200 Hz) traces of the membrane potential of excitatory neurons of the CA1 network (light blue) with detected SWRs (stars). (**C**) Distribution of thalamic slow wave troughs (down states) locked to the cortical slow wave trough (vertical dashed line at zero) occurring within ±0.5 s plotted in 20ms bins in the model. The arrow indicates the median of the distribution (n=277, median ± MAD = 0.113±0.019 s). Thalamic slow waves occur significantly later than cortical slow waves (p<10⁻²⁰, Wilcoxon signed rank test). (**D**) Distribution of thalamic spindle onsets locked to the cortical spindle onset (vertical dashed line at zero) occurring within ±2 s plotted in 100ms bins for one session. The arrow indicates the

*Figure 5 continued on next page*

*Figure 5 continued*

median of the distribution (n=211, median ± MAD = –0.102±0.387 s). Thalamic spindles occur significantly earlier than cortical spindles (p=1.30 × 10⁻⁷, Wilcoxon signed rank test). (E) Phase-amplitude coupling (PAC) of slow waves and spindles, calculated from the average time-frequency representation of amplitudes in the spindle frequency range, locked to the trough of the mPFC slow wave. Left, average of cortical slow wave trough-locked time-frequency representations (n=391) of the membrane potential of mPFC excitatory neurons (top) and reuniens neurons (bottom). Black curves represent grand average filtered membrane potential of cortical excitatory neurons in the slow wave frequency range (0.5–2 Hz) aligned to the cortical slow wave trough (time 0). Right, circular histogram of synchronization index (SI) angles of cortical (top) and reuniens (bottom) spindles relative to the cortical slow waves. SI angles were nonuniformly distributed (mean ± SD = 148.7±23.9°, p<10⁻²⁰ and mean ± SD = 27.4±20.4°, p<10⁻²⁰, for mPFC and reuniens respectively, Rayleigh test, n=391 slow waves). (F) Histogram of the slow wave–spindle coupling strengths (the absolute value of the synchronization index) for mPFC (blue) and reuniens (green) spindles. Vertical lines show the median of the distribution (p=8.45 × 10⁻⁹, Wilcoxon signed rank test, median ± MAD SI strength = 0.58 ± 0.11 and 0.66±0.15 for mPFC and reuniens, respectively). (G) Average filtered membrane potential of cortical, reuniens and CA1 excitatory neurons, in the slow wave frequency range (0.5–2 Hz), aligned to the cortical slow wave trough (time 0). (H) Distribution of slow wave phases at the time of the SWRs. Zero phase shows the peak (up state) of slow waves (n=714, mean ± SD = –24.97±62.01°, p<10⁻²⁰, Rayleigh test). (I) Average membrane potential of the mPFC, reuniens and CA1 excitatory neurons aligned to the SWR peaks. (J) Average membrane potential of the mPFC (top) and reuniens (middle) neurons and histogram of SWR incidence (bottom), aligned to the onset of cortical spindles. *p<0.05, **p<0.01, ***p<0.001.

thalamocortical circuitry between the mPFC and reuniens. Slow waves in reuniens lagged prefrontal slow waves but spindles in reuniens led prefrontal spindles, suggesting a bidirectional, oscillation-dependent dialogue between reuniens and the mPFC. Spindle amplitude and SWR occurrence was modulated by the phase of the prefrontal slow oscillation, consistent with the idea that prefrontal slow waves control the timing of thalamic and hippocampal events. Overall, the study provides physiological confirmation to the hypothesis that reuniens is a core mediator in hippocampo-cortical dialogue.

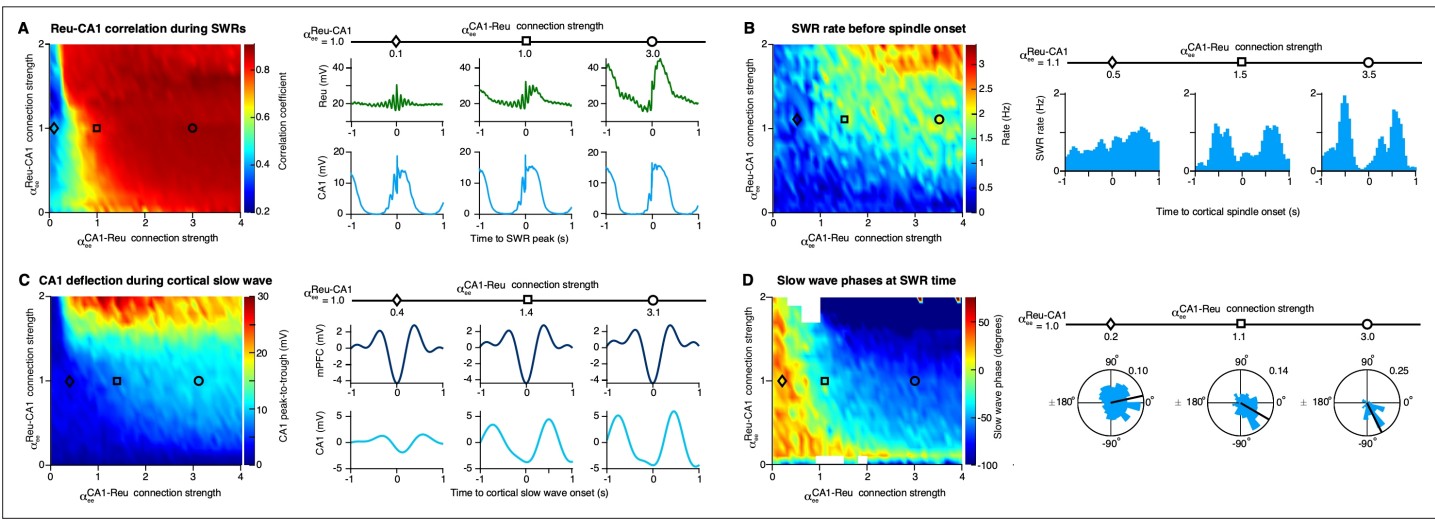

**Figure 6.** Bidirectional projections between the reuniens and CA1 control multiregional interactions. (A) Left, correlation between reuniens and CA1 membrane potentials during SWRs (maximum value of the covariance between reuniens and CA1 membrane potentials ± 0.5 s around the SWR peak) versus the CA1-reuniens and reuniens-CA1 connection strengths ($\alpha_{ee}^{CA1-Reu}$ and $\alpha_{ee}^{Reu-CA1}$ show the factor multiplied to $J_{ee}^{CA1-Reu}$ and $J_{ee}^{Reu-CA1}$ representing the strength of the input from CA1 to reuniens and from reuniens to excitatory neurons of CA1 respectively. $\alpha_{ee}^{CA1-Reu} = \alpha_{ee}^{Reu-CA1} = 1$ corresponds to the network shown in *Figure 5*). Right, average traces of the reuniens and CA1 membrane potentials aligned to the SWR peaks for three different values of CA1-reuniens connection strengths. (B) Left, SWR rate preceding the cortical spindle onset (peak-to-trough of the histogram of SWR incidence in a window of –1 s before the spindle onset) versus the CA1-reuniens and reuniens-CA1 connection strengths. Right, histogram of SWR incidence time-locked to the onset of cortical spindles for three different values of CA1-reuniens projection strengths. (C) Left, peak-to-trough of the CA1 deflections around cortical slow wave trough (±1 s) versus the CA1-reuniens and reuniens-CA1 connection strengths. Right, average filtered membrane potential of cortical, and CA1 excitatory neurons, in the slow wave frequency range (0.5–2 Hz) aligned to the cortical slow wave trough (time 0) for three different values of CA1-reuniens connection strengths. (D) Left, mean slow wave phase at the time of SWRs versus the CA1-reuniens and reuniens-CA1 connection strengths. Mean phases are shown only for bins with nonuniformly distributed of phases (p<0.05, Rayleigh test) and omitted for uniformly distributed phase (white boxes). Right, histogram of slow wave phases at the time of SWRs for 3 different values of CA1-reuniens connection strengths, showing decreasing mean phase with increasing CA1-reuniens connections strength (p=4.28 x 10⁻⁸, p=7.53 x 10⁻¹⁵ and p=2.53 x 10⁻³⁹ for $\alpha_{ee}^{CA1-Reu}$ = 0.2, 1.1 and 3.0, respectively).

## Monosynaptic and polysynaptic hippocampal to mPFC connections

Hippocampal influence on the activity of prefrontal neurons is a central mechanism in sleep-dependent memory consolidation (*Bontempi et al., 1999*; *Siapas et al., 2005*; *Wilson and McNaughton, 1994*). Previous research has shown that response latency in the hippocampo-prefrontal pathway exhibits substantial variability (*Dégenètais et al., 2003*), indicating the involvement of both monosynaptic and polysynaptic mechanisms in prefrontal response to hippocampal input. Our findings support this notion, as we observed inconsistent mPFC responses to hippocampal stimulation, occurring later than those elicited by reuniens stimulation. Notably, the same cell in mPFC could respond or not respond to hippocampal stimuli (*Figure 1C and E*) while LFP responses were less variable. This was possibly due to inhibitory mechanisms such as shunting/feedforward inhibition or cholinergic, monoaminergic, or thalamic gating mechanisms (*Floresco and Grace, 2003*; *Gigg et al., 1994*; *Gioanni et al., 1999*; *Mansvelder et al., 2009*; *Robbins, 2005*; *Williams and Goldman-Rakic, 1995*).

## Reuniens to mPFC pathway induces more reliable responses than hippocampus to mPFC pathway

While the functional impact of the hippocampo-prefrontal pathway has been extensively studied, there are fewer studies on the effects of reuniens on the prefrontal cortex (*Di Prisco and Vertes, 2006*). In our study, half of recorded prefrontal cells responded to reuniens stimulation, with a mean response latency of 6ms and with antidromic spikes with a latency of approximately 3ms, pointing to bidirectional functional connectivity between the two structures. Similarly, electrical stimulation of the reuniens/rhomboid nucleus in rats (*Di Prisco and Vertes, 2006*) elicits prominent evoked potentials in prefrontal cortices that have putatively monosynaptic latencies of 4.5ms. We found that mPFC responses to reuniens stimulation were qualitatively different from responses elicited by hippocampal stimulation, with significant differences in latency, amplitude, slope of EPSPs, and duration of hyperpolarization. Reuniens-evoked responses occurred earlier and were more consistent than hippocampal-evoked responses, suggesting higher response fidelity in the reuniens-prefrontal pathway than in the hippocampo-cortical pathway. Both pyramidal cells and inhibitory neurons mediate the hippocampo-prefrontal response, with the early component thought to be primarily mediated by AMPA receptors (*Dégenètais et al., 2003*; *Jay et al., 1992*) and the early hyperpolarization was likely mediated by $GABA_A$ and the late component by either $GABA_B$ conductances (*Dégenètais et al., 2003*) or it was mediated by disfacilitation (*Contreras et al., 1996*; *Fuentealba et al., 2004*). mPFC interneurons also respond to hippocampal stimulation, with a shorter response latency than pyramidal cells, suggesting that the hippocampus can exert feedforward inhibition in the medial prefrontal cortex (*Tierney et al., 2004*), which could explain the polarity reversal of mPFC neurons response around –70 mV to hippocampal stimulation (*Figure 1E*). AMPA excitation and GABA inhibition may also mediate the primary depolarizing and secondary hyperpolarizing responses to reuniens stimulation in medial prefrontal cells (*Figure 1C and F*) similar to previously demonstrated mediodorsal-prefrontal pathway (*Gigg et al., 1992*). Similar to hippocampal afferents to the mPFC, the reuniens nucleus sends glutamatergic afferents to the prefrontal cortex (*Hur and Zaborszky, 2005*; *Pirot et al., 1995*) enabling influence on the prefrontal neuronal activity.

## A potential role of Reuniens to synchronize mPFC and hippocampal activity

Reuniens is believed to play a critical role in connecting the prefrontal cortex and hippocampus due to its bidirectional connections with both structures (*Di Prisco and Vertes, 2006*). The prefrontal cortex and hippocampus are both essential for memory processing (*Bontempi et al., 1999*), and previous research indicates the reuniens may synchronize the activity of these two structures (*Ferraris et al., 2018*). Demonstrations that the nucleus reuniens exerts excitatory actions at the CA1 (*Bertram and Zhang, 1999*; *Dolleman-Van der Weel et al., 1997*) and also at the prefrontal cortex (*Di Prisco and Vertes, 2006*; intracellularly confirmed and described in our study) provides supporting evidence for this claim that nucleus Reuniens might synchronize the activity of ventral hippocampus and mPFC. We also found that hippocampal stimulation evoked strong responses in cortical areas coincident with the recording site of antidromic responses to reuniens stimulation (*Figure 1*). These results support the idea that the reuniens nucleus, ventral hippocampus, and prefrontal cortex form a functional loop, with the prefrontal cortex acting as a mediator of the hippocampo-prefronto-thalamic relay.

In NREM sleep, the coupling of hippocampal SWRs to cortical slow waves and thalamocortical spindles is thought to be the main network mechanisms by which the brain promotes the consolidation of short-term memory into long-term storage (*Frankland and Bontempi, 2005*; *Maingret et al., 2016*). However, the mechanism that governs the emergence of this interregional synchrony remains unclear. We found that reuniens single-unit activity exhibited sporadic locking to ventral hippocampal theta cycles (during REM sleep) and was strongly modulated by ventral hippocampal SWRs. These results are congruent with findings in MD thalamic nucleus showing that although overall CA1 ripples suppress MD firing, at the onset of spindles, the MD firing triggered by ripples was increased (*Yang et al., 2019*). The firing pause in reuniens neurons during SWR and the presence of spike-bursts after SWR suggests that the spike burst is a rebound from previous hyperpolarization (*Jahnsen and Llinás, 1984*). This suggests that SWR drive some inhibitory cells that hyperpolarize at least MD and reuniens. If hyperpolarization is strong enough, like during slow wave, it triggers rebound spike bursts. Considering the strong influence of reuniens on prefrontal targets, the hippocampal modulation of reuniens activity points to a thalamic pathway for the coupling of sleep rhythms. Although the minimal requirement to generate spindles is the reticular thalamic nucleus (TRN; *Bazhenov et al., 1999*; *Steriade et al., 1987*), our finding of increased SWR probability and ripple power prior to the onset of thalamocortical spindles suggest that hippocampal drive through reuniens may initiate the thalamocortical cascade of events that leads to spindles.

## Does Reuniens generate spindles?

In first-order nuclei of the thalamus, cortical firing can trigger spindles by exciting hyperpolarized neurons of the TRN, eliciting high-frequency spike bursts (*Contreras and Steriade, 1996*). In turn, TRN bursts can drive IPSPs in thalamocortical neurons, eliciting rebound low-threshold calcium spikes (*Bazhenov et al., 1999*) and developing the spindle primarily through recurrent inhibitory-excitatory interactions between the TRN and thalamocortical neurons (*Contreras and Steriade, 1996*). Cortical firing also contributes to spindle synchronization and the maintenance of the spindle through its influence on the TRN (*Bonjean et al., 2011*). Thus, cortical activation after the down state can initiate thalamocortical spindling and lock the timing of spindles to the phase of the slow oscillation. However, less is known about spindle generation in higher-order thalamic nuclei, especially in thalamic nuclei with limbic circuitry such as reuniens. The circuitry is in place for spindle generation in the reuniens nucleus: The reuniens is connected bidirectionally with the prefrontal cortex (*Herkenham, 1978*; *Vertes et al., 2007*); layer 6 of the infralimbic cortex projects to the anteromedial TRN (*Cornwall et al., 1990*), a region that projects to the reuniens nucleus (*Cavdar et al., 2008*; *Kolmac and Mitrofanis, 1997*). Our results revealed that reuniens neuronal firing is modulated by spindles, whether spindles were recorded by LFP electrodes located in the mPFC cortex or in the reuniens nucleus (*Figure 2*). While reuniens nucleus neurons can generate low-threshold calcium spike, they do not display hyperpolarization-activated current (*Walsh et al., 2017*), likely limiting their ability to generate and maintain spindles. Of note, without MD in our computational model, no spindles were generated (data not shown). In general, although first order thalamocortical cells display inhibitory, spindle-like activities during cortical active state, higher order thalamocortical cells show persistent depolarizing states, excluding their possibility to generate spindle-related rebound spike bursts (*Groh et al., 2014*; *Sheroziya and Timofeev, 2014*). The bidirectional connections of the reuniens to neocortical and archicortical structures make it difficult to discern the possible triggering event that initiates the spindle in this nucleus. We found that hippocampal SWRs trigger thalamic rebound burst firing and precede the onset of reuniens and mPFC spindles, which points to SWRs as one of candidate events for spindle initiation. In this regard, the strong excitatory drive emerging form the hippocampus during SWRs may represent an ontogenetically different mechanism for the initiation of spindles in the limbic circuitry. However, the generation and maintenance of spindle cycles depend on burst firing from inhibitory structures such as the TRN. To date, it is not yet clear whether the hippocampus influences inhibitory thalamic structures such as the TRN or the zona incerta, although projections from subiculum to the rostral TRN were previously described (*Cavdar et al., 2008*). We propose that hippocampal SWRs can initiate a cascade of thalamocortical events, beginning with the excitation of reuniens, which in turn excites the mPFC. This mPFC excitation leads to spindle activity generated through cortico-thalamic feedback from the mPFC to other thalamic nuclei, such as the MD nucleus.

## Our computational model recaptures our experimental findings

Our computational model demonstrates the presence of cross-regional interactions between the prefrontal cortex and its thalamic partners. The model confirmed that thalamic slow waves occur significantly later than cortical slow waves, whereas thalamic spindles occur significantly earlier than cortical spindles, recapturing our experimental findings and pointing to the cortical origin of slow-wave activity and thalamic origin of spindles. The model predicts that, while CA1 to reuniens connection strength plays a critical role in the modulation of reuniens activity by SWRs, bidirectional connections between reuniens and CA1 are critical for SWR-dependent triggering of spindles. Similarly, bidirectional reuniens and CA1 connections are important for transmitting cortical slow wave activity to the CA1 region. With strong bidirectional connections, the SWR occurred primarily before the slow wave active phase peak and with weak connections, the SWR tended to occur after this peak. Given our experimental findings (*Figure 3J and K*) the model suggests that the effectiveness of bidirectional connections between reuniens and CA1 is particularly strong during NREM in cat.

## Conclusion

Overall, in addition to well-known anatomical connectivity in CA1, reuniens, mPFC network, our results demonstrate the presence of functional connectivity in this network. However, the efficacy of this functional connectivity is modulated by behavioral state. Such flexibility can be explained by neuromodulation of intrinsic neuronal currents, synaptic strength and synaptic dynamic (*Nadim and Bucher, 2014*).

## Materials and methods

### Key resources table

| Reagent type (species) or resource | Designation | Source or reference | Identifiers | Additional information |
|---|---|---|---|---|
| Strain, cat (males and females) | Cat | Marshall BioResources | Marshall Cat, domestic short-hair | |
| Software, algorithm | MATLAB | This paper, copy archived at *Basha, 2024* | | https://github.com/DiellorBasha/MATLAB_for_Electrophysiology |
| Software, algorithm | MATLAB | This paper, copy archived at *Azarmehri, 2024* | | https://github.com/azarmehri/brain-network-interactions |
| Software, algorithm | Spike2 9.14 | Cambridge Electronic Design | RRID:SCR_000903 | |
| Software, algorithm | FieldTrip | *Oostenveld et al., 2011* | RRID:SCR_004849 | |
| Software, algorithm | CircStat | *Berens, 2009* | RRID:SCR_016651 | |
| Software, algorithm | UltraMegaSort2000 (ums2k) | Neurophysics Lab, UCSD | RRID:SCR_015857 | https://neurophysics.ucsd.edu/software.php |
| Software, algorithm | Freely Moving Animal (FMA) Toolbox | | RRID:SCR_015533 | https://fmatoolbox.sourceforge.net/ |
| Software, algorithm | LabChart 8 | ADI Instruments | RRID:SCR_023643 | |
| Software, algorithm | IgorPro 7 | Wavemetrics | N/A | |
| Software, algorithm | Canvas 11 | ACDSee | RRID:SCR_014288 | |
| Chemical compound, drug | DAPI | Invitrogen | D3571 | |
| Other, equipment | AM 3000 amplifiers | A-M systems | N/A | |
| Other, equipment | PowerLab 35 Series Data Acquisition | ADI instruments | N/A | |
| Other, equipment | Neurodata IR-283 amplifiers | Cygnus Technology | N/A | |

*Continued on next page*

*Continued*

| Reagent type (species) or resource | Designation | Source or reference | Identifiers | Additional information |
|---|---|---|---|---|
| Other, equipment | Tetrodes | Custom, This paper | N/A | |

## Experimental model and subject details

Experiments were carried out in accordance with the guidelines of the Canadian Council on Animal Care and were approved by the Committee for Animal Care of Université Laval (protocols 2020–663 and 2016–104) and are consistent with ARRIVE guidelines. In 20 adult cats (16 anesthetized [acute, results in *Figure 1*] during recording sessions and 4 chronically implanted, not anesthetized [chronic, results in *Figures 2–4*] during recording sessions), aged 1–1.5 years, electrophysiological recordings were obtained from the mPFC, midline thalamus, hippocampus and/or various cortical areas including somatosensory, motor, auditory, associative, and visual cortex. Prefrontal intracellular recordings (n=15) and LFP-array recordings (n=9) were obtained from anesthetized animals and spike/local field potential (n=4) recordings were obtained from non-anesthetized animals during sleep and wake cycles.

## Surgeries

### Anesthetized animals

For acute intracellular recordings, cats were anesthetized with a mixture of ketamine and xylazine (10–15 and 2–3 mg/kg i.m., respectively) and maintained in anesthetic state during the experiment by supplemental doses of ketamine (5 mg/kg). Body temperature was maintained at 37 °C via a feedback-controlled heating pad and the heart rate was continuously monitored (90–110 bpm). All pressure points and incision sites were infiltrated with lidocaine (0.5%). Animals were paralyzed with gallamine triethiodide and artificially ventilated to an end-tidal $CO_2$ concentration of 3.5–3.8%. Recording stability was maintained by cannulation of the cisterna magna, bilateral pneumothorax, hip suspension, and by sealing craniotomies with agar solution (4% in 0.9% saline).

### Non-anesthetized animals

For chronic spike/LFP recordings, cats were implanted with indwelling electrodes and a head-restraint system according to surgical procedures described previously in *Timofeev et al., 2001* and *Chauvette et al., 2011*. Briefly, cats were pre-anesthetized with an intramuscular injection of ketamine (3 mg/kg), buprenorphine (0.02 mg/kg), and dexmedetomidine (5 µg/kg). The incision site was shaved, and the cats were intubated for gaseous anesthesia. Lidocaine (0.5%) and bupivacaine (0.25%) was injected at the site of the incision and in all pressure points where the head contacted the stereotaxic frame. Four stainless-steel electrodes were implanted under stereotaxic guidance in the mPFC (*from the interaural line: AP 23–24 mm, ML 1 mm, DV 5 mm, θ=10°*) according to the Reinoso-Suarez stereotaxic atlas of the cat brain (*Reinoso-Suárez, 1961*). Additionally, single stainless-steel electrodes were implanted in deep cortical layers (1.2 mm from cortical surface) in the ipsi- or contralateral hemisphere in the marginal gyrus [visual cortex (areas 17 and 18)], suprasylvian gyrus [associative cortex (areas 5, 7, and 21)], ectosylvian gyrus [auditory cortex (areas 22 and 50)], postcruciate gyrus [somatosensory cortex (area 3)], precruciate gyrus [motor cortex (areas 4 and 6)]. Tetrodes, constructed out of four twisted microwires (platinum-iridium, 12.5 µm, gold-plated to 190–210 kΩ), were implanted into the midline thalamus (*from the interaural line: AP 11.5, ML 0.5, DV 1.5, θ=10°*) and into the CA1 region of the hippocampus (*from the interaural line: AP 7, ML 10.5, DV –3.5, θ=10°*).

For electro-oculography (EOG), a silver electrode was implanted through the orbital bone and for electromyography (EMG), two teflon-insulated stainless-steel electrodes were implanted into the neck muscle. All recordings were referenced to a silver electrode fixed to the skull above the cerebellum. To allow for subsequent head-restrained recordings, head-fixation holders were also attached and encased into the dental cement. Throughout the surgery, the body temperature was maintained at 37 °C using a water-circulating thermo-regulated blanket. Heartbeat, oxygen saturation and blood pressure were continuously monitored using a pulse oximeter (Rad-8; MatVet), and the level of anesthesia was adjusted to maintain a heartbeat at 130–150 per minute. Lactate Ringer's solution was

given intravenously (5 ml/ kg/h, i.v.) during the surgery. After the surgery, cats received meloxicam (0.05 mg/kg) once a day for 3 days.

## Electrophysiological recordings

### Acute recordings in anesthetized animals

Intracellular recordings of the mPFC were obtained in vivo using glass micropipettes pulled on a vertical puller (Narishige PP-830, Tokyo, Japan) from borosilicate capillary tubes (WPI; P-97, Sutter Instrument) and filled with 3 M potassium acetate (DC resistances of 30–60 MΩ). The prefrontal region around the cruciate gyrus was targeted according to the stereotaxic cat atlas of *Reinoso-Suárez, 1961* (*from the interaural line: AP 23–24 mm, ML 1 mm, DV 5–14.5 mm*). The micropipette was advanced dorsoventrally until a cell was penetrated, confirmed in situ by electrophysiological observations and classification. A high-impedance amplifier (Neurodata IR-283 amplifiers; Cygnus Technology) with active bridge circuitry was used to record the membrane potential (sampled at 20 kHz) and to inject current into neurons.

Stimuli (0.2ms, 0.3–1.5 mA) were delivered in single pulses at 1 Hz through bipolar (coaxial) electrodes implanted in the ventral midline thalamus (*from the interaural line: AP 12, ML 0.0, DV 1.0*) and hippocampus (*from the interaural line: AP 7.2, ML 10.5, DV –4.5*). Bandpassed (0.1 Hz-10 kHz) field potentials were recorded through the same bipolar electrodes at 20 kHz sampling.

To study the topography of prefrontal responses to thalamic and/or hippocampal stimulation, an array of four tungsten microelectrodes (9–12 MΩ, 1 mm apart in the antero-posterior axis) was positioned in the mPFC in nine animals (ML 1.0, AP 26–23). The array was positioned close to the midline in the pericruciate gyrus sigmoidus and advanced ventrally in 1 mm steps. At each step, thalamic or hippocampal stimuli (0.2ms, 0.3–1.5 mA) were delivered at 1 Hz or in 5-pulse trains every 2 s at 5 Hz, 10 Hz, 20 Hz and 100 Hz through bipolar (coaxial) electrodes implanted in the ventral midline thalamus (*from the interaural line: AP 12, ML 0.0, DV 1.0*) and hippocampal formation (*from the interaural line: AP 7.2, ML 10.5, DV –4.5*). Antidromic events were detected according to methods described previously in *Lipski, 1981*.

To label stimulation sites, electrolytic lesions were made through thalamic and hippocampal electrodes via 0.75 mA, 1 s current pulses, delivered every 3 s for 5 min.

### Chronic recordings in non-anesthetized animals

For non-anesthetized cats, a post-operative recovery time of 1 week was maintained prior to the first recording session. Cats were trained over 2–3 days to remain in head-restrained position for 2–4 hr and cycle through periods of quiet wakefulness, NREM, and REM sleep. Recordings were obtained over two weeks following the post-operative recovery period. All recordings were conducted within a Faraday chamber using AM 3000 amplifiers (A-M Systems), bandpassed 0.1 Hz to 10 kHz with a 1 k gain. All signals were sampled at 20 kHz and digitized with PowerLab (ADInstruments - Data Acquisition Systems for Life Science, RRID:SCR_001620).

## Histology

Location of all electrodes was verified on histological sections from each animal. At the conclusion of experiments, animals were deeply anesthetized and perfused intracardially with 4 °C saline (0.9%) followed by 4% paraformaldehyde (PFA) in 0.1 M phosphate buffer in saline (PBS). Brains were extracted and cryoprotected by immersion in several sucrose-PFA solutions of increasing sucrose concentration (10, 20 and 30% sucrose in PFA over 1 week). Brains were then sectioned in a freezing microtome (80 or 100 µm thickness), mounted on gelatinized glass slides, dried, and subsequently labelled with DAPI (300 nM, Invitrogen D3571) or stained with cresyl violet by standard Nissl-staining procedures. The coverslipped sections were visualized in brightfield (Nissl) or fluorescence (DAPI) microscopy and sections with electrode tracks were photographed in ×10 and ×20 magnification.

## Analysis

Signals were analyzed using Igor Pro 4.0 (Wavemetrics, Inc), built-in and custom routines in Spike2 (Cambridge Electronic Design, Ltd.) and custom-written and open-source code in MATLAB (Mathworks Inc).

## Intracellular data analysis

Stimuli were detected manually, and minimal binomial smoothing was performed to diminish noise on occasion. The latency of orthodromic responses was calculated as the time when the response reached 10% of the maximal amplitude of the sigmoidal fit of the rise of the EPSP. The amplitude of the EPSP was calculated as the maximum of this sigmoidal fit, and the slope of the EPSP calculated as the maximum of the differentiation of this fit. A horizontal line was drawn across the response from the base of the early EPSP and the interception of this line by the hyperpolarizing potential was used to calculate the duration of this latter component. Responses to sweeps of five pulse stimulation trains were averaged with an IgorPro algorithm (n=30 per stimulation protocol). The EPSP response magnitude elicited by the stimulation of the reuniens nucleus was computed by computing the area below the curve. Average response areas evoked by pulses 2 through 5 of the train were subsequently normalized by dividing by the response evoked by the first pulse.

## Wake, NREM and REM detection

Detection of wake, NREM and REM sleep was based on automated analysis of the mPFC and EMG signals, followed by manual REM scoring using mPFC, EMG, and EOG data. For automated detection, the Spike2 script RatSleepAuto, based on rodent recordings (*Costa-Miserachs et al., 2003*), was adapted to cat mPFC recordings. Briefly, the mPFC signal was downsampled to 100 Hz and bandpassed in the delta range (1–4 Hz). The mPFC, EMG and the delta-filtered mPFC signals were then squared and the mean of each signal was calculated in 5 s bins. Each 5 s bin was classified as NREM if the mPFC and delta signals exceed the mean + 1 SD of the signal and if the EMG fell below mean - 1 s.d. All bins other than NREM were initially classified as wake. In the subsequent manual correction, 'wake' bins with EMG below mean – 1 s.d. and EOG above mean +1 s.d. were classified as REM. In the final stage, the 5 s bins were combined into groups of 4 to create 20 s epochs of wake, REM or NREM according to the majority of 5 s bins that comprised the 20 secondepoch. Epochs without a majority of either NREM, REM or wake were labelled as 'Transition'.

## Slow wave, spindle and SWR detection

Slow wave, spindle and SWR detection was based on methodology described previously in *Alizadeh et al., 2022* and respectively in *Zugaro et al., 2022*, *Klinzing et al., 2016* and *Levenstein et al., 2019*. The detection protocol for all three events was restricted to periods of NREM sleep.

The Freely Moving Animal (FMA) toolbox (*Zugaro et al., 2022*) was used for slow wave and spindle detection. For slow waves, mPFC and reuniens LFP signals were bandpass filtered between 0.5–4 Hz using a FIR filter with a filter order corresponding to 3 cycles of the low frequency cut-off. Tentative slow wave events whose positive peaks were greater than 2.5 standard deviations of the signal and whose difference between positive and negative peaks was greater than 4 standard deviations were detected. Slow waves were finally defined as periods when consecutive positive-to-negative zero crossings of the signal filtered in 0.5–2 Hz occurred between 0.5 and 2 s of each other.

For spindles, mPFC and reuniens LFP signals were bandpassed in the spindle frequency band (7–15 Hz) using finite-impulse-response (FIR) filters from the EEGLAB toolbox (*Delorme and Makeig, 2004*). The instantaneous amplitude of the filtered signal was then computed from its Hilbert transform and the results were smoothed using a 300ms Gaussian window. Potential spindles were detected as periods when this signal crossed 3 SD above its mean for 0.5–3 s A threshold of 2.5 SD above the mean was set to detect spindle onset and offset. Events that occurred within 0.5 s of each other were merged into a single event. Events longer than 0.5 s and shorter than 3 s were defined as spindles. The largest peak in spindle amplitude was defined as the time of the spindle event.

For SWRs, hippocampal LFP signals were bandpassed in the ripple frequency band (150–200 Hz, FIR filter from the EEGLAB toolbox). The root mean square (RMS) of the filtered signal was then computed and the results were smoothed using a 50ms Gaussian windows. SWRs were defined as periods when the smoothed RMS crossed 4 SD above the median and stayed crossed for at least 30ms. The largest peak in SWR amplitude was defined as the time of the SWR event.

## Phase-amplitude coupling

Computations of phase-amplitude coupling (PAC) were based on methodology published previously in *Alizadeh et al., 2022*, *Dehnavi et al., 2021* and *Staresina et al., 2015*. Time-frequency representations (TFRs) of every spindle/slow wave event were obtained using the *mtmconvol* function in the FieldTrip toolbox (*Oostenveld et al., 2011*). Spindle power around the slow wave was estimated using sliding Hanning tapered windows (10ms steps) with a variable length that included five cycles. To compute coupling, TFRs were then normalized as percentage change from pre-event baseline (−3.5 to −2.5 s before event) for all events. To calculate the modulation of spindle power by slow wave phase, TFR bins around slow waves were averaged across the spindle frequency range and then filtered in the slow wave frequency range. Phase values of this signal were calculated from their Hilbert transform.

## Synchronization index and coupling strength

We defined a synchronization index (SI) for each event:

$$SI = \frac{1}{m} \sum_{j=1}^{m} e^{i\left[\theta_{slowwave}(j) - \theta_{spindle}(j)\right]}$$

where m is the number of time points, $\theta_{slowwave}(j)$ represents slow wave phase value at time *j* and $\theta_{spindle}(j)$ represents phase values of spindle power fluctuations at time *j*. Preferred phase was estimated in the interval –0.25–0.25 s around slow wave peak. The strength of phase-amplitude coupling was estimated from the absolute value of the SI with 1 representing the most consistent phase shift between spindle amplitude and slow wave event. The phase of SI represents the amount of phase shift between the spindle amplitude and slow wave event. Spindle-SWR coupling was computed using a similar procedure. Briefly, the normalized TFRs in the SWR frequency range around each spindle event (baseline defined from –2.5 to –1.5 s before spindle event) were filtered in the spindle frequency range and the SI was defined as the vector mean of unit vectors, each showing the phase difference between SWR amplitude and slow wave event at different time points around the spindle peak.

## Average wavelet transforms

Wavelet transforms of the LFP were calculated using the continuous one-dimensional wavelet transform, estimated by Morse wavelets defined in the *cwt* function, available in the MATLAB wavelet toolbox. Average wavelet transforms around spindle onsets (*Figure 2L*) were computed using custom-written MATLAB code. First, 5 s LFP epochs, centered on spindle onsets were extracted from the continuous recordings and downsampled to 250 Hz. A filter bank, based on the known epoch length (5 s) and sampling frequency (250 Hz), was created with voices per octave set to 20 and frequency limits set to 0.1–50 Hz. Using the filter bank, wavelet transforms of each epoch around a spindle onset were calculated and stored in a three-dimensional array corresponding to time, frequency, and spindle event. To obtain the magnitude of the signal, the absolute value of the wavelet transform was then computed. To obtain the average wavelet transform of the LFP signals around the spindle event onsets, the mean of the magnitude matrices along the third dimension (spindle events) was calculated.

## Spike detection

For spike detection, reuniens LFP signals were bandpass filtered in the 300–8000 Hz range using finite impulse response (FIR) filters from the EEGLAB toolbox. Epochs of ± 250ms around times when the signal crossed ±16 SD were considered as artifacts and removed from the analysis. Spikes were detected using UltraMegaSort2000 (*Fee et al., 1996*) with a negative threshold of –4SD. During a shadow time of 0.75ms after the threshold crossing, no new event was detected to avoid multiple threshold crossings of one event. A window of 1.5ms was considered around each detected spike from 0.6ms before the threshold crossing time to 0.9ms after this time.

   Spike sorting was based on a method described previously in *Souza et al., 2019*. We first extracted spike features for each channel. The features consisted of peak-to-trough value, trough value and also an index which was set to one for the channel with maximum trough amplitude and zero for the other channels. We used principal component analysis (PCA) to combine 12 features (3 features times 4 channels) and selected the two first components. Gaussian distributions were fitted to PCA

components with gaussian mixture model (GMM) algorithm. Each distribution was determined as a potential spike cluster. We further limited the distribution by removing spikes with large Mahalanobis distance from the center of the distribution. We removed, combined, and limited the distributions manually by visual inspection of the plots showing PCA components versus each other and the plots showing the features for different channel pairs (*Figure 2—figure supplement 1*). For multi-unit analysis, spikes were detected according to threshold crossing and template matching algorithms in Spike2 (*CED, 2021*). Briefly, fast signals (~1–3ms) that exceeded the threshold were first detected as putative spikes and timestamped. Thresholds were manually determined according to the signal-to-noise ratio of spikes in each recording. Next, a temporary spike template was formed according to the shape of the putative spike and a 'template width' was estimated (twice the mean difference between the template and the spike that created it). Each new spike was then compared against the template and added to it if the spike's sample points fell within the template width. The template was modified with the addition of each new spike up to a maximum of eight spikes after which the template was checked against previous templates. If a match was found, the temporary template was merged with the existing template. Otherwise, a new confirmed template was generated.

## Peri-event histograms

To analyze reuniens single-unit activity around the onset of SWRs, peri-event spike histograms (PETHs) were calculated for a 1 second period around each event onset and the results were smoothed with a 10ms Gaussian window. For comparison, surrogate PETHs were obtained by randomly shuffling SWR events detected within 1 second windows. This procedure was repeated 100 times. True-even PETHs were z-score normalized to the firing rate of surrogate PETHs. Single-units with activity greater than 2 SDs and smaller than −2 SDs in the 150ms window following SWR onset were considered activated and suppressed, respectively.

## Phase-locking analysis

For phase-locking analysis, we calculated spindle phase at the time of single-unit spikes occurring within ±0.25 s around spindle peak (*Figure 2H*), slow wave phase at SWR time (*Figure 3K*) and theta phase at single-unit spike time occurring within ±0.5 s around theta peak (*Figure 4C*). First, mPFC and reuniens LFPs were filtered in the frequency band of interest (spindle, slow wave, or theta) and the phase of the filtered signal was extracted from its Hilbert transform. These phase values were used to obtain phase histograms and calculate circular means.

## Computational model

Our model represents a minimal neural mass model that describes the network architecture required for generating three key oscillatory activities during NREM: the slow oscillation, spindles, and SWRs. The hippocampal-thalamocortical model represents CA1-CA3 networks, two cortical networks (CX, CX'), and three thalamic networks consisting of mediodorsal nucleus of the thalamus (MD), thalamic reticular nucleus (TRN) and nucleus reuniens (REU). The details of the CA1-CA3, MD-TRN and cortical networks were described previously (*Azimi et al., 2021*; *Ghorbani et al., 2012*; *Hashemi et al., 2019*). In brief, in this neural mass firing rate model, each of cortical and hippocampal networks consists of one identical group of excitatory and one identical group of inhibitory neurons. $V_k^m$ and $N_k^m$ indicate the membrane potential and the number of type k (either e, excitatory or i, inhibitory) neurons of network m, respectively. The firing rates of hippocampal and cortical neurons as well as the reuniens neurons are sigmoid functions of the membrane potential of the neurons with the sharpness and threshold of $g_k^m$ , $V_k^{*m}$ respectively:

$$r(V)_k^m = r_0 + \frac{r_1}{1 + exp[-(V_k^m - V_k^{*m})/g_k^m]}, m \in CA3, CA1, CX, CX', REU \tag{1}$$

where, $r_1$ = 70 Hz ($r_0$=0.1 Hz) is the maximal (minimum) firing rate. T-type calcium currents is modeled only for MD and TRN neurons by defining a bursting variable, $u_m$ so that these neurons show burst mode in addition to the tonic mode:

$$r(V_k^m) = \frac{R_k^T}{1 + exp[(V_m^k - V_k^T)/g_k^T]}(exp[L_k u_k]) + \frac{R_k^B}{1 + exp[(V_m^k - V_k^B)/g_k^B]}(1 - exp[L_k u_k]), m \in MD, TRN \tag{2}$$

Here $g_k^T$ ($g_k^B$), $R_k^T$ ($R_k^B$), and $V_k^T$($V_k^B$) show the sharpness of the firing rate dependence on the membrane potential, the maximum firing rate, and the threshold potential during tonic (burst) mode. The sharpness of switch between the two modes is controlled by $L_k$ (a positive parameter).

The probability and the strength of connections from neuron k of network m to neuron j of network n are shown by $P_{kj}^{m-n}$ and $J_{kj}^{m-n}$, respectively. m and n can be CA1, CA3, CX, MD, TRN, or REU. k and j can take e or i (representing excitatory and inhibitory neurons, respectively). For the short-range connections since m and n are equal, only one of them is shown. E-E connections of cortical and hippocampal excitatory neurons are subject to dendritic spike frequency adaptation so that $J_{ee}^m/J_{ee}^{n-m}$ is a sigmoid function of an adaptation variable, $c^m$ with sharpness and threshold of $g_c$ and $c^*$, respectively:

$$J_{ee}^m(c) = \frac{J0_{ee}^m}{1 + exp[c - c^*/g_c]} \tag{3}$$

Here $J0_{ee}^m$ is the maximal synaptic strength. The full model is described by 17 rate equations:

$$\frac{dV_e^{CA3}}{dt} = -\frac{V_e^{CA3}}{\tau_e} + N_e^{CA3}P_{ee}^{CA3}J_{ee}^{CA3}(c^{CA3})r(V_i^{CA3}) - N_i^{CA3}P_{ie}^{CA3}J_{ie}^{CA3}r(V_i^{CA3}) \tag{4}$$

$$\frac{dV_i^{CA3}}{dt} = -\frac{V_i^{CA3}}{\tau_i} + N_e^{CA3}P_{ei}^{CA3}J_{ei}^{CA3}r(V_e^{CA3}) - N_i^{CA3}P_{ii}^{CA3}J_{ii}^{CA3}r(V_i^{CA3}) \tag{5}$$

$$\frac{dc^{CA3}}{dt} = -\frac{c^{CA3}}{\tau_c} + N_e^{CA3}P_{ee}^{CA3}\Delta c^{CA3}r(V_e^{CA3}) \tag{6}$$

$$\begin{aligned}\frac{dV_e^{CA1}}{dt} &= -\frac{V_e^{CA1}}{\tau_e} - N_i^{CA1}P_{ie}^{CA1}J_{ie}^{CA1}r(V_i^{CA1}) \\ &+ N_e^{CA3}P_{ee}^{CA3-CA1}J_{ee}^{CA3-CA1}(c^{CA1})r(V_e^{CA3}) \\ &+ N_e^{REU}P_{ee}^{REU-CA1}J_{ee}^{REU-CA1}(c^{CA1})r(V_e^{REU})\end{aligned} \tag{7}$$

$$\begin{aligned}\frac{dV_i^{CA1}}{dt} &= -\frac{V_i^{CA1}}{\tau l} + N_e^{CA1}P_{ei}^{CA1}J_{ei}^{CA1}r(V_e^{CA1}) + N_e^{CA3}P_{ei}^{CA3-CA1}J_{ei}^{CA3-CA1}r(V_e^{CA3}) - \\ &N_i^{CA1}P_{ii}^{CA1}J_{ii}^{CA1}r(V_i^{CA1}) + N_e^{REU}P_{ei}^{REU-CA1}J_{ei}^{REU-CA1}r(V_e^{REU})\end{aligned} \tag{8}$$

$$\frac{dc^{CA1}}{dt} = -\frac{c^{CA1}}{\tau_c} + N_e^{CA3}P_{ee}^{CA3-CA1}\Delta c^{CA1}r(V_e^{CA3}) + N_e^{REU}P_{ee}^{REU-CA1}\Delta c^{CA1}r(V_e^{REU}) \tag{9}$$

$$\begin{aligned}\frac{dV_e^{CX}}{dt} &= -\frac{V_e^{CX}}{\tau_e} + N_e^{CX}P_{ee}^{CX}J_{ee}^{CX}(c^{CX})r(V_e^{CX}) - N_i^{CX}P_{ie}^{CX}J_{ie}^{CX}r(V_e^{CX}) + \\ &N_e^{'CX}P_{ee}^{CX2-CX1}J_{ee}^{CX2-CX1}(c^{CX})r(V_e^{'CX}) + N_e^{REU}P_{ee}^{REU-CX}(c^{CX})r(V_e^{REU}) + N_e^{CA1}P_{ee}^{CA1-CX}J_{ee}^{CA1-CX}(C^{CX})r(V_e^{CA1}\end{aligned} \tag{10}$$

$$\begin{aligned}\frac{dc^{CX}}{dt} &= -\frac{c^{CX}}{\tau_c} + N_e^{CX}P_{ee}^{CX}\Delta c^{CX}r(V_e^{CX}) + N_e^{'CX}P_{ee}^{CX2-CX1}\Delta c^{CX}r(V_e^{'CX}) + \\ &N_e^{REU}P_{ee}^{REU-CX}\Delta c^{CX}r(V_e^{REU}) \\ &+ N_e^{CA1}P_{ee}^{CA1-CX}(\Delta c^{CX})r(V_e^{CA1})\end{aligned} \tag{11}$$

$$\begin{aligned}\frac{dV_i^{CX}}{dt} &= -\frac{V_i^{CX}}{\tau l}N_i^{CX}P_{ii}^{CX}J_{ii}^{CX}r(V_i^{CX}) + N_e^{CX}P_{ei}^{CX}J_{ei}^{CX}r(V_e^{CX}) + N_e^{'CX}P_{ei}^{CX2-CX1}J_{ei}^{CX2-CX1}r(V_e^{'CX}) \\ &+ N_e^{REU}P_{ee}^{REU-CX}r(V_e^{REU}) + N_e^{CA1}P_{ei}^{CA1-CX}J_{ei}^{CA1-CX}r(V_e^{CA1})\end{aligned} \tag{12}$$

$$\begin{aligned}\frac{dV_i^{'CX}}{dt} &= -\frac{V_e^{'CX}}{\tau l} - N_i^{'CX}P_{ie}^{'CX}J_{ie}^{CX}r(V_l^{'CX}) + N_e^{'CX}P_{ee}^{'CX}J_{ee}^{'CX}r(V_e^{'CX}) + N_e^{CX}P_{ee}^{CX1-CX2}J_{ee}^{CX1-CX2}r(V_e^{'CX}) \\ &+ N_e^{'MD}P_{ee}^{'MD-CX}J_{ee}^{'MD-CX}(c^{'CX})r(V_e^{MD})\end{aligned} \tag{13}$$

$$\frac{dc^{'CX}}{dt} = -\frac{c^{'CX}}{\tau_c} + N_e^{'CX}P_{ee}^{'CX}\Delta c^{'CX}r(V_e^{'CX}) + N_e^{CX}P_{ee}^{CX1-CX2}\Delta c^{'CX}r(V_e^{CX}) + N_e^{MD}P_{ee}^{'MD-CX}\Delta c^{'CX}r(V_e^{MD}) \tag{14}$$

$$\begin{aligned}\frac{dV_l^{'CX}}{dt} &= -\frac{V_l^{'CX}}{\tau l} + N_e^{'CX}P_{ei}^{'CX}J_{ei}^{'CX}r(V_e^{'CX}) - N_I^{'CX}P_{ii}^{'CX}J_{ii}^{'CX}r(V_e^{'CX}) + N_e^{CX}P_{ei}^{CX1-CX2}J_{ei}^{CX1-CX2}r(V_e^{CX}) + \\ &N_e^{MD}P_{ei}^{'MD-CX}J_{ei}^{'MD-CX}r(V_e^{MD})\end{aligned} \tag{15}$$

$$\frac{dV_e^{MD}}{dt} = -\frac{V_e^{MD}}{\tau_{MD}} - \frac{f_e^{max}}{1 + exp[\frac{u_e + f_e^{th}}{q_e}]} - N_i^{TRN}P_{ie}^{TRN-MD}J_{ie}^{TRN-MD}r(V_i^{TRN}) + N_e^{'CX}P_{ee}^{'CX-MD}J_{ee}^{CX-MD}r(V_3^{'CX})$$

(16)

$$\frac{du_e}{dt} = [\frac{1}{\tau_e^u}(b_e - u_e)] \ \text{if} \ V_e^{MD} > -0.1 \ mv \ \text{then} \ b_e = 0 \ \text{(otherwise)} \ b_e = -200 \ mA \tag{17}$$

$$\frac{dV_i^{TRN}}{dt} = -\frac{V_i^{TRN}}{\tau_{TRN}} - \frac{f_i^{max}}{1 + exp[\frac{u_I + f_i^{th}}{q_i}]} - N_i^{TRN}P_{ii}^{TRN}J_{ii}^{TRN}r(V_i^{TRN}) + N_e^{MD}P_{ei}^{MD-TRN}J_{ei}^{MD-TRN}r(V_e^{MD}) +$$

$$N_e^{'CX}P_{ei}^{'CX-TRN}J_{ei}^{'CX-TRN}r(V_e^{'CX'})$$

(18)

$$\frac{du_i}{dt} = [\frac{1}{\tau_i^u}(b_I - u_I)] \ \text{if} \ V_i^{TRM} > 0 \ mv \ \text{then} \ b_I = 0 \ \text{otherwise} \ b_I = -200 \ mA \tag{19}$$

$$\frac{dV_e^{REU}}{dt} = -\frac{V_e^{REU}}{\tau^{REU}} - N_i^{TRN}P_{ie}^{TRN-REU}J_{ie}^{TRN-REU}r(V_i^{TRN}) + N_e^{CX}P_{ee}^{CX-REU}r(V_e^{CX}) + N_e^{CA1}P_{ee}^{CA1-REU}J_{ee}^{CA1-REU}$$

(20)

The nonlinear equations of the model were solved using the fourth-order Runge–Kutta method with a time step of 1ms by MATLAB R2017b software.

## Quantification and statistical analysis

Statistical analyses were performed using the MATLAB Statistics and Machine Learning Toolbox (Mathworks Inc, Natick, MA, USA). Phase analysis was performed using the circular statistics tool box developed by *Berens, 2009*. The Rayleigh test was performed to determine the uniformity in the phase preference of events. The Watson-Williams test was used to test the inequality of phases. Polar plots showing phases, and mean phases were plotted using MATLAB.

Intracellular data was analyzed using JMP 5.01 software. Normality was assessed by normal quantile plots and by the Shapiro-Wilk test. Homogeneity of variance was assessed by Levene's test. Confidence intervals were set at 95% (p$P$<0.05). ANOVA was used when data was normally distributed and variance homogeneous, Welch ANOVA when variance was non-homogeneous. To determine group differences in normally distributed data, Tukey's range test was used. Otherwise, differences were assessed by non-parametric testing (van der Waerden test). Standard deviations are reported as SD and the mean absolute deviation is reported as MAD.

## Acknowledgements

We thank Serge Ftomov for technical assistance.

# Additional information

## Funding

| Funder | Grant reference number | Author |
| --- | --- | --- |
| Natural Sciences and Engineering Research Council of Canada | RGPIN-2018-06291 | Igor Timofeev |
| Canadian Institutes of Health Research | PTJ - 183862 | Igor Timofeev |

The funders had no role in study design, data collection and interpretation, or the decision to submit the work for publication.

## Author contributions

Diellor Basha, Conceptualization, Data curation, Software, Formal analysis, Investigation, Visualization, Methodology, Writing – original draft; Amirmohammad Azarmehri, Software, Formal analysis,

Visualization, Methodology, Writing – original draft; Elian Proulx, Data curation, Formal analysis, Investigation, Visualization, Methodology, Writing – original draft; Sylvain Chauvette, Resources, Validation, Investigation, Methodology, Writing - review and editing; Maryam Ghorbani, Conceptualization, Supervision, Validation, Investigation, Methodology, Project administration, Writing - review and editing; Igor Timofeev, Conceptualization, Data curation, Supervision, Funding acquisition, Validation, Investigation, Visualization, Methodology, Project administration, Writing - review and editing

#### Author ORCIDs
Diellor Basha https://orcid.org/0000-0001-9312-8633
Maryam Ghorbani https://orcid.org/0000-0001-8671-6728
Igor Timofeev https://orcid.org/0000-0002-1389-5857

#### Ethics
Experiments were carried out in accordance with the guidelines of the Canadian Council on Animal Care and were approved by the Committee for Animal Care of Université Laval (protocols 2020-663 and 2016-104) and are consistent with ARRIVE guidelines.

Reviewer #1 (Public review): https://doi.org/10.7554/eLife.90826.3.sa1
Reviewer #2 (Public review): https://doi.org/10.7554/eLife.90826.3.sa2
Author response https://doi.org/10.7554/eLife.90826.3.sa3

## Additional files

#### Supplementary files
MDAR checklist

#### Data availability
The following data sets were generated Basha, Diellor; Timofeev, Igor (2022), "Natural sleep electrophysiology: mPFC, thalamus, hippocampus", Mendeley Data, V2, doi: 10.17632/4x27jjp9mv.2 (Basha and Timofeev, 2022). MATLAB code generated for this study is available at https://github.com/DiellorBasha/MATLAB_for_Electrophysiology/tree/main/eLife2025_12_RP90826 and https://github.com/azarmehri/brain-network-interactions (copy archived at *Basha, 2024* and *Azarmehri, 2024*). Tetrodes and stainless-steel electrodes manufactured in-house for this study are available from the lead contact with a completed materials transfer agreement. This study did not generate new unique reagents. Further information and requests for resources and reagents should be directed to and will be fulfilled by the Lead Contact, Igor Timofeev (igor.timofeev@fmed.ulaval.ca).

The following dataset was generated:

| Author(s) | Year | Dataset title | Dataset URL | Database and Identifier |
|---|---|---|---|---|
| Basha D, Timofeev L | 2022 | Natural sleep electrophysiology: mPFC, thalamus, hippocampus | http://doi.org/10.17632/4x27jjp9mv.2 | Mendeley Data, 10.17632/4x27jjp9mv.2 |

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
