## [Editor Report · eLife Assessment]

The **important** manuscript presents **convincing** evidence of temporal correlations during specific oscillatory activity between the prefrontal cortex, thalamic nucleus reuniens, and the hippocampus, in naturally sleeping animals. Such correlations represent **solid** evidence to support the notion that the thalamic nucleus reuniens participates in the hippocampal and prefrontal cortex dialogue subserving memory processes.

---

## [Referee Report · Reviewer #1 (Public review)]

Summary:

In this study, Basha and colleagues aim to test whether the thalamic nucleus reuniens can facilitate the hippocampus/prefrontal cortex coupling during sleep. Considering the importance of sleep in memory consolidation, this study is important to understand the functional interaction between these three majorly involved regions. This work suggests that the thalamic nucleus reuniens has a functional role in synchronizing the hippocampus and prefrontal cortex. Therefore, it paves the way to new perspectives in order to decipher the neuronal and circuit mechanisms underlying such processes.

Strengths:

The authors have used an interdisciplinary approach to determine how the thalamic nucleus reuniens can impact on the cortico-hippocampal dialogue during sleep. They performed recording in naturally sleeping cats, and analysed the correlation between the main slow wave sleep oscillatory hallmarks: slow waves, spindles, and hippocampal ripples, and with reuniens' neurons firing. They also associated intracellular recordings to assess the reuniens-prefrontal connectivity, and computational models of large networks in which they determine that the coupling of oscillations is modulated by the strength of hippocampal-thalamic connections.

Since the literature regarding fundamentals in nucleus reuniens anatomy in cats is much thinner as compared to what is available in rodents, the authors have performed complementary functional anatomy experiments in anesthetised cats in order to support the functional results (aka excitatory links between reuniens and its targets).

Weaknesses:

The authors have used cats as animal models to study the hippocampo-cortical dialogue. Whereas this is a very interesting and well-done study, it will address a more limited audience, as the majority of research is conducted on rodents, which have a different functional anatomy. Hence, the mechanisms of triggering of thalamic spindles would therefore be far less relevant in rodents, unless shown otherwise in the future.

---

## [Referee Report · Reviewer #2 (Public review)]

Summary:

The interplay between the medial prefrontal cortex and ventral hippocampal system is critical for many cognitive processes, including memory and its consolidation over time. A prominent idea in recent research is that this relationship is mediated at least in part by the midline nucleus reuniens with respect to consolidation in particular. Whereas the bulk of evidence has focused on neuroanatomy and the effects of temproary or permanent lesions of the nucleus reuniens, the current work examined the electrophysiology of these three structures and how they inter-relate, especially during sleep, which is anticipated to be critical for consolidation. They provide evidence from intercellualr recordings of the bi-directional functional connectivity among these structures. There is an emphasis on the interactions between these regions during sleep, especially slow wave sleep. They provide evidence, in cats, that cortical slow waves precede reuniens slow waves and hippocampal sharp-wave ripples, which may reflect prefrontal control of the timing of thalamic and hippocampal events, They also find evidence that hippocampal sharp wave ripples trigger thalamic firing and precede the onset of reuniens and medial prefrontal cortex spindles. The authors suggest that the effectiveness of bidirectional connections between the reuniens and the (ventral) CA1 is particularly strong during non-rapid eye movement sleep in the cat. This is a very interesting, complex study on a highly topical subject.

Strengths:

An excellent array of different electrophysiological techniques and analyses are conducted. The temporal relationships described are novel findings that suggest mechanisms behind the interactions between the key regions of interest. These may be of value for future experimental studies to test more directly their association with memory consolidation.

Weaknesses:

The number of findings provided is complex and some readers may struggle to follow all the details. The fact that bidirectional connections exist in the model system is not new per se. How and why the specific findings add to existing literature could still be presented with a little more impact. However, I am not sure this can be done more easily than is currently presented. I leave that to the authors to consider.

---

## [Author Response]

The following is the authors’ response to the original reviews.

**Public Reviews:**

**Reviewer #1 (Public Review):**
Summary:In this study, Basha and colleagues aim to test whether the thalamic nucleus reuniens can facilitate the hippocampus/prefrontal cortex coupling during sleep. Considering the importance of sleep in memory consolidation, this study is important to understand the functional interaction between these three majorly involved regions. This work suggests that the thalamic nucleus reuniens has a functional role in synchronizing the hippocampus and prefrontal cortex.Strengths:The authors performed recordings in naturally sleeping cats, and analysed the correlation between the main slow wave sleep oscillatory hallmarks: slow waves, spindles, and hippocampal ripples, and with reuniens' neurons firing. They also associated intracellular recordings to assess the reuniens-prefrontal connectivity, and computational models of large networks in which they determined that the coupling of oscillations is modulated by the strength of hippocampal-thalamic connections.

Thank you for your positive evaluation.

Weaknesses:The authors' main claim is made on slow waves and spindle coupling, which are recorded both in the prefrontal cortex and surprisingly in reuniens. Known to be generated in the cortex by cortico-thalamic mechanisms, the slow waves and spindles recorded in reuniens show no evidence of local generation in the reuniens, which is not anatomically equipped to generate such activities. Until shown differently, these oscillations recorded in reuniens are most likely volume-conducted from nearby cortices. Therefore, such a caveat is a major obstacle to analysing their correlation (in time or frequency domains) with oscillations in other regions.

(1) We fully agree with the reviewer that reuniens likely does not generate neither slow waves nor spindles. We do not make such claim, which we clearly stated in the discussion (lines 319-324). We propose that Reuniens neurons mediate different forms of activity. In the model, we introduced MD nucleus only because without MD we were unable to generate spindles. While the slow waves and spindles are generated in other thalamocortical regions, the REU neurons show these rhythms due to long-range projections from these regions to REU as has been shown in the model.

(2) Definitely, we cannot exclude some influence of volume conductance on obtained LFP recordings in REU nucleus. However, we show modulation of spiking activity within REU by spindles. Spike modulation cannot be explained by volume conductance but can be explained by either synaptic drive (likely the case here) or some intrinsic neuronal processes (like T-current).

(3) In our REU recordings for spike identification we used tetrode recordings. If slow waves and spindles are volume conducted, then slow waves and spindles recorded with tetrodes should have identical shape. Following reviewer comment, we took these recordings and subtracted one channel from another. The difference in signal during slow waves is in the order 0.1 mV. Considering that the distance between electrodes is in the order of 20 um, such a difference in voltage is major and can only be explained by local extracellular currents, likely due to synaptic activities originating in afferent structures.

Finally, the choice of the animal model (cats) is the best suited one, as too few data, particularly anatomical ones regarding reuniens connectivity, are available to support functional results.

(1) Thalamus of majority of mammals (definitely primates and carnivores, including cats) contain local circuit interneurons (about 30 % of all neurons). A vast majority of studies in rodents (except LGN nucleus) report either absence or extremally low (i.e. Jager P, Moore G, Calpin P, et al. Dual midbrain and forebrain origins of thalamic inhibitory interneurons. eLife. 2021; 10: e59272.) number of thalamic interneurons. Therefore, studies on other species than rodents are necessary, and bring new information, which is impossible to obtain in rodents.

(2) Cats’ brain is much larger than the brain of mice or rats, therefore, the effects of volume conductance from cortex to REU are much smaller, if not negligible. The distance between REU and closest cortical structure (ectosylvian gyrus) in cats is about 15 mm.

(3) Indeed, there is much less anatomical data on cats as opposed to rodents. This is why, we performed experiments shown in the figure 1. This figure contains functional anatomy data. Antidromic responses show that recorded structure projects to stimulated structure. Orthodromic responses show that stimulated structure projects to recorded structure.

**Reviewer #2 (Public Review):**
Summary:The interplay between the medial prefrontal cortex and ventral hippocampal system is critical for many cognitive processes, including memory and its consolidation over time. A prominent idea in recent research is that this relationship is mediated at least in part by the midline nucleus reuniens with respect to consolidation in particular. Whereas the bulk of evidence has focused on neuroanatomy and the effects of temproary or permanent lesions of the nucleus reuniens, the current work examined the electrophysiology of these three structures and how they inter-relate, especially during sleep, which is anticipated to be critical for consolidation. They provide evidence from intercellular recordings of the bi-directional functional connectivity among these structures. There is an emphasis on the interactions between these regions during sleep, especially slow-wave sleep. They provide evidence, in cats, that cortical slow waves precede reuniens slow waves and hippocampal sharp-wave ripples, which may reflect prefrontal control of the timing of thalamic and hippocampal events, They also find evidence that hippocampal sharp wave ripples trigger thalamic firing and precede the onset of reuniens and medial prefrontal cortex spindles. The authors suggest that the effectiveness of bidirectional connections between the reuniens and the (ventral) CA1 is particularly strong during non-rapid eye movement sleep in the cat. This is a very interesting, complex study on a highly topical subject.Strengths:An excellent array of different electrophysiological techniques and analyses are conducted. The temporal relationships described are novel findings that suggest mechanisms behind the interactions between the key regions of interest. These may be of value for future experimental studies to test more directly their association with memory consolidation.

We thank this reviewer for very positive evaluation of our study.

Weaknesses:Given the complexity and number of findings provided, clearer explanation(s) and organisation that directed the specific value and importance of different findings would improve the paper. Most readers may then find it easier to follow the specific relevance of key approaches and findings and their emphasis. For example, the fact that bidirectional connections exist in the model system is not new per se. How and why the specific findings add to existing literature would have more impact if this information was addressed more directly in the written text and in the figure legends.

Thank you for this comment. In the revised version, we will do our best to simplify presentation and more clearly explain our findings.

**Reviewing Editor (Recommendations for Authors):**
Please discuss the ability of reuniens to generate spindles?

We briefly discussed this in previous version. We now extended the discussion (p. 18).

For population data, how many cats were used in acute and chronic experiments, where does the population data originate in Fig. 2? How repeatable were the findings across animals? Was histology verified in each animal?

As previously stated in the beginning of method section we totally used 20 cats: 16 anesthetized (or acute) and 4 non-anesthetized (or chronic). We added number of cats in appropriate places in the result section. Population data in figure 2 comes from 48, 49 or 52 recording sessions (depending on the type of analysis, and indicated in the figure legend) from 4 chronic cats; we clarified this information in the legend. Results were highly repeatable across animals. Histology was verified in all chronic and acute animals, we added a sentence in the method section.

Explanation of figures is very poor, values in figures should be reported in results so they can be compared in the context of the description.

In this revised version, we report most numbers present in figures and their legend to the main text (result section).

The depth of the recording tungsten electrodes are meaningless without the AP and ML coordinates given how heterogenous mPFC is. What is the ventromedial wall of the mPFC in the cat?

We added the ML and AP coordinates in the method section. We corrected ventromedial wall for ventroposterior part of the mPFC.

What are the two vertical lines in 1F?

This was an error while preparing the figure. The panel was corrected.

Line 90 mean +-SD of what? There are no numbers.

Thanks, we now indicate the values.

Panel 2L does not show increased spindling in reuniens prior to PFC as indicated in the results, please explain. It does show SWR in the hippocampus prior to spindles, what is the meaning of such a time relationship?

Panel 2L did show an increased spindling reuniens prior to mPFC, but indeed at the time scale shown, it was not very clear. In this revised manuscript, we added an inset zooming around time zero to make this point clearer.

Panel 2L indeed show an increase in SWR prior to the increase in spindle in both Reuniens and mPFC.

As stated in the discussion, ‘We found that hippocampal SWRs trigger thalamic firing and precede the onset of reuniens and mPFC spindles, which points to SWRs as one of candidate events for spindle initiation.’

It is unclear what the slow waves of PFC mean, these represent filtered PFC lfp, but is this a particular oscillation? They continue to occur during the spindle, while the slow waves supposedly trigger the spindle. Please explain and clarify.

We recently published a review article involving several scientists studying both human and animal sleep that has inserted Box. 1 (Timofeev I, Schoch S, LeBourgeois M, Huber R, Riedner B, Kurth S. Spatio-temporal properties of sleep slow waves and implications for development. Current Opinion in Physiology. 2020; 15: 172–182). In this box among other terms, we provide current definition of slow waves vs slow oscillation. Briefly, if slow waves are repeated with a given rhythm, they typically form slow oscillation. However, if they occur in isolation or are not rhythmic, they remain slow waves, but cannot be called slow oscillation.

Regarding relation of spindles and slow oscillation. We are currently systematically analyzing data on spindles and slow waves obtained from head-restrained and freely behaving cats. One of the main findings is that a majority of ‘cortical’ spindles are local. Local to the extent that spindles can occur in alternation in two neighboring cortical cells. Largely, LFP sleep spindles occur more or less synchronously within suprasylvian gyrus of cats where indeed a large majority of them was triggered by slow waves. The synchrony between LFP spindles in suprasylvian vs other other cortical areas is much less clear. So, it is not surprizing that spindles in one bran region can occur when there is a slow wave present in some other brain region. Something of a kind was also shown in human (Mölle M, Bergmann TO, Marshall L, Born J. Fast and slow spindles during the sleep slow oscillation: disparate coalescence and engagement in memory processing. Sleep. 2011; 34 (10): 1411-1421).

In this regard, we are not ready to include modifications in the manuscript.

Line 134, where is spindle amplitude shown? Plots report power within the spindle frequency band, which obviously captures more than just spindles.

No, plots of figure 3 B, C show the phase-amplitude coupling (PAC) strength. These were calculated with detected spindles, therefore, while we cannot exclude some false spindle detections, we are confident that the false spindle detections are at a negligible level. We modified text and instead of spindle amplitude, we describe SW-spindle amplitude coupling. This reflects our analysis with exactitude.

The discussion must include the medio dorsal nucleus which is the largest thalamic input to the prefrontal cortex and also receives input from the hippocampus. In particular, the case must be made for why reuniens would play a more important or different role than MD? (For example: Occurrence of Hippocampal Ripples is Associated with Activity Suppression in the Mediodorsal Thalamic Nucleus - PMC (nih.gov)).

We cited the suggested study. We cannot say whether reuniens plays a more or less important role. What is clear is that hippocampal ripples at the onset of spindles trigger increased firing in both MD and reuniens. Our extracellular recordings (Fig. 4, K) suggest that the increased firing is associated with spike-bursts. We also have a parallel unpublished study done on anesthetized mice showing SWR triggered inhibitory potentials in both reuniens and MD that reverses around -65mV - -70 mV. Because the majority of SWR occurred at the onset of cortical up state, a relative role of cortico-thalamic vs hippocampo-thalamic drive is not easy to separate. We hope, we will convincingly do this in our forthcoming study, with the limitation that it was done on anesthetized mice.

**Reviewer #1 (Recommendations For The Authors):**
I strongly encourage the authors to perform current source density analyses on the LFP signals recorded in the nucleus reuniens to make sure that the observed oscillations are indeed locally generated. So far, the anatomical organisation in reuniens cannot support the local generation of oscillations, such as spindles and slow wave. At least in rodents (the cat reuniens does not seem too different, until shown differently), there were no oscillators found in reuniens, and at least not arranged like in cortical areas, allowing the summation in time, and particularly space, of rhythmic input currents. Bipolar recordings with pairs of twisted electrodes might also be useful to assess the local existence of spindles and slow waves.

Current source density calculation is possible when one knows the exact distance between recording sites. As we used tetrodes made with 4 twisted platinum-iridium wires, we know more or less the range of distance between recording sites, but not the exact distance between any given pair of electrodes.

Then, the physical distance between the reuniens and any cortical structure is about 8-9 mm. Therefore, with such distances, volume conductance is expected to be negligible. If slow waves and spindles are volume conducted, then slow waves and spindles recorded with tetrodes should have identical shape. Following reviewer comment, we took these recordings and subtracted one channel from another. The difference in signal during slow waves is in the order 0.1 mV. Considering that the distance between electrodes is in the order of 20 um, such a difference in voltage is major and can only be explained by local extracellular currents, likely due to synaptic activities originating in afferent structures.

Below, we plotted the voltage of one channel of the tetrode versus another channel of the same tetrode. If the signal was simply volume conducted, one would expect to see the vast majority of points on the x=y line (red).

Below is a segment of mPFC LFP recording (upper black trace), mPFC LFP filtered for spindle frequency (7-15 Hz) and the spindle detected black lines above the filtered trace. Then two LFP traces from a tetrode in the Reuniens (orange and light blue) are overlayed. The second trace (Blue) from bottom represents the substraction of Reuniens 1 minus Reuniens 2 channel, and just below (lower Blue trace) is this susbtraction trace filtered for spindle frequency (7-15 Hz) showing clear voltage difference in the spindle range between the two electrodes. Note also that around time 179-179.5 s, there is clear spindle oscillation in the mPFC recording which is not present in the Reuniens recordings.

**Author response image 2. sa3fig2:** 

Therefore, we are convinced that in our recordings, volume conductance did not play any significant role.

Another concern regarding delays between events, like slow waves, measured between two regions (as exemplified by Figure 3). It appears that the delays were calculated from the filtered signal. Figure 3G shows a delay between the peak of the mPFC slow wave between the raw and the filtered signal, which might be artifactual of the processing. It is though not (or less) visible for the reuniens recording. Such mismatch might explain the observed differences in delays.

Thanks for this comment. We recomputed the analysis using the original signal (smoothed) and obtained very similar results. Panels H and I of figure 3 were updated using the new analysis performed on original signal.

The overall analyses of LFP-triggered reuniens MUA activity lack of statistics (at least z-scored firing to normalise the firings).

Fig. 2 H and I are representative examples for histograms; statistical data are shown in circular plots as explained in the legend. Fig. 2 L, shows populational data and we provide now standard error. Fig. 4 C and D show individual example. Fig. 4 E shows histograms of activity of all identified putative single units. Units that show significant modulation are displayed above white line. Fig. 4 F shows populational data for significantly modified units.

A last point of detail in the model, which surprisingly shows reuniens to excitatory hippocampal cells' connectivity. Recent literature reports that reuniens only connect hippocampal interneurons, and not principal cells (at least in rodents, I could not find any report in cats). I wonder how changing this parameter would affect the results of the computational investigation, particularly the results shown in Figure 6.

There are several studies in the literature showing a direct excitation from the Reuniens to pyramidal cells in the CA1, here are three of them:

Goswamee, P., et al. (2021). "Nucleus Reuniens Afferents in Hippocampus Modulate CA1 Network Function via Monosynaptic Excitation and Polysynaptic Inhibition." Frontiers in Cellular Neuroscience 15.

Dolleman-Van der Weel MJ, Lopes da Silva FH, Witter MP (1997) Nucleus Reuniens Thalami Modulates Activity in Hippocampal Field CA1 through Excitatory and Inhibitory Mechanisms. The Journal of Neuroscience 17:5640.

Dolleman-van der Weel MJ, Lopes da Silva FH, Witter MP (2017) Interaction of nucleus reuniens and entorhinal cortex projections in hippocampal field CA1 of the rat. Brain Structure and Function 222:2421-2438.

Because this is not a review paper, we opted to not cite all the papers describing connectivity between mPFC, hippocampus and thalamus.

**Reviewer #2 (Recommendations For The Authors):**
I respectively suggest that the earlier (public) comments listed above should be addressed. In addition, it would be useful to make it clearer when non-rapid eye movement sleep was being addressed and when rapid eye movement was being addressed. Is it of value to use a single term instead of adding "slow wave sleep" or else clarify when either term is used? The addition of more subheadings might help. Moreover, the relative contribution/value of evidence from these two sleep states was not addressed or was not very clear.

We tried to make it clearer when NREM and when REM was analysed.

We replaced slow-wave sleep with NREM sleep in the figure 5 title.

We added several subheadings in the discussion.

Relative contribution of NREM vs REM sleep was not addressed? Sorry but we do not clearly understand your question. Figs. 2 and 3 deal mainly with NREM sleep (Fig 2.B has an example of REM sleep). Fig. 4 essentially describes results obtained during REM sleep.

I was not sure if the Abstract summarised the key take-home messages from the large amount of evidence provided. Some choices are needed, of course, but "evidence of bidirectional connectivity" struck me as less novel than other evidence provided. Given the huge amount of findings provided, which is commendable, it is still useful to present it perhaps in a more digestible fashion. For example, the headings or the first sentence(s) below headings could indicate the aim or the outcome of the specific method/analysis/findings.

We rewrote abstract and we also added some conclusion to highlight major findings and their meaning.

It is more common to use NRe or Re, rather than REU.

We avoided using RE as, for decades, we used RE to abbreviate the thalamic reticular nucleus in several publications. In this revised version, we spell at full - Reuniens.

Line 49 mentions "short-term" memory. Please specify this more clearly as it is otherwise ambiguous. Also, line 303.

We rephrased the sentence: In particular, the hierarchical coupling of slow waves, spindles and SWRs is thought to play a key role in memory consolidation.

Line 303 was likely about the ventromedial wall: we corrected that sentence.

Line 62: the word, "required" (for memory function) is too strong because there is evidence that it is not always required.

We modified the sentence for plays a major role.

The focus within the medial prefrontal cortex could be specified more clearly / earlier.

The mPFC is mentioned in the second sentence of the abstract and in the first sentence of the introduction.

Line 134: The heading states "determine" and then mentions modulation. These terms may not be interchangeable or they need clarification.

We changed it to slow wave-spindle amplitude coupling. This represents exactly our analysis.

Line 204: Does "cortical network" mean prefrontal cortex network"?

Yes, as described in lines 192-193, the two cortical networks (N1 and N2) of the model represent the mPFC layer 5 and 6 respectively.

Lines 283 to 289: These were not very clear to me.

These lines described the potential mechanisms for the responses to hippocampal and reuniens stimulation recorded intracellularly (results in figure 1). We modified this paragraph for clarity.

Line 296: Specify the "claim".

We modified the sentence for “[…] provides supporting evidence for this claim that nucleus Reuniens might synchronize the activity of ventral hippocampus and mPFC.”

The discussion naturally focuses on the thalamic nucleus reuniens, but also occasionally mentions the thalamic mediodorsal nucleus. The distinction, assuming this is highly relevant, could be expressed more clearly (direct comparison with their previous papers).

We never published a study on the mediodorsal nucleus. We do have some unpublished results from recordings in the MD nucleus and they reveal the presence of an inhibitory component at the beginning of cortical active states, therefore behaving in a similar way to first order nuclei. It is then possible that spindles recorded in the reuniens are actually generated in the MD nucleus and then transmitted to Reuniens through the thalamic reticular nucleus, as both MD and reuniens are connected to the rostral thalamic reticular nucleus. We added some discussion about this.

Figure 1B: Do the authors have any additional evidence of the placements in the reuniens, because the photo provided suggests a large area beyond the reuniens boundary. Also, please confirm is the CEM between Rh and Re in the cat (I think the Rh and Re are adjacent in the rat).

Figure 1B is from an electrolytic lesion, which is necessarily bigger than the tip of the electrode. Therefore the center of the electrolytic lesion indicates where the electrode tip was located which is well within the reuniens nucleus.

Also, yes CE (Nucleus centralis thalami, pars medialis) is located between the reuniens and rhomboid in cats. This can be found in two cat atlas:

Reinoso-Suárez, F. (1961). Topographischer Hirnatlas der Katze für experimental-physiologische Untersuchungen (Merck).

Berman AL, Jones EG (1982) The Thalamus and Basal Telencephalon of the Cat: A Cytoarchitectonic Atlas with Stereotaxic Coordinates: University of Wisconsin Press.

The first mention of hippocampus in the figure legends should remind the reader by stating "ventral hippocampus".

In this revised version, we added “ventral” in several instances both in the main text and in figure legend.

Figure 2: It seems unusual to mention "unusually short NREM". Presumably, things are the same otherwise - if so, perhaps mention that, especially if some of the effects reflect an "unusual" episode.

We display this particular segment because we want to show continuous recording in which still individual elements characterizing specific states are still visible.

Some effects look like they are strong and others perhaps weaker. If so, how do these impact the final conclusions?

Sorry, we did not understand clearly what is meant here by the reviewer. In general, if any effect has statistically significant difference (old fashion 0.05) we consider it as significant. Any other cases are described on individual basis.

Perhaps "MAD" should be in full on the first occasion, if not already.

It was spelled out at line 659, but we now spell it out also in the results section and in figure 2 legend.

Methods: the key question is the use of rodent recordings to classify cat recordings. It would be good to have a reference indicating that this can be directly used for cats, which may have different sleep cycles and patterns compared to rats.

We did not use rodent recordings to classify cat recordings, however we did used a state detection script that was developed with rodent recordings. As mentioned in the method section, we adapted the script to cat mPFC recordings and then manual corrections were made to correctly detect REM episodes. Respectfully, our lab investigates sleep-wake in non-anesthetized animals for a few decades; we developed state detection algorithm in mice, cats, marmosets when needed (to analyse months of recordings), and we have an extensive expertise in identifying states of vigilance from electrophysiological recordings.